# APOBEC3B-mediated corruption of the tumor cell immunopeptidome induces heteroclitic neoepitopes for cancer immunotherapy

Christopher B. Driscoll[1,2,13], Matthew R. Schuelke [1,3,4,13], Timothy Kottke[1], Jill M. Thompson[1], Phonphimon Wongthida[1], Jason M. Tonne[1], Amanda L. Huff[1,2], Amber Miller[1], Kevin G. Shim[1,3,4], Amy Molan[5], Cynthia Wetmore[6], Peter Selby[7], Adel Samson[7], Kevin Harrington[8], Hardev Pandha[9], Alan Melcher[10], Jose S. Pulido[11], Reuben Harris [5], Laura Evgin[1,14] & Richard G. Vile[1,3,12,14]*

APOBEC3B, an anti-viral cytidine deaminase which induces DNA mutations, has been implicated as a mediator of cancer evolution and therapeutic resistance. Mutational plasticity also drives generation of neoepitopes, which prime anti-tumor T cells. Here, we show that overexpression of APOBEC3B in tumors increases resistance to chemotherapy, but simultaneously heightens sensitivity to immune checkpoint blockade in a murine model of melanoma. However, in the vaccine setting, APOBEC3B-mediated mutations reproducibly generate heteroclitic neoepitopes in vaccine cells which activate de novo T cell responses. These cross react against parental, unmodified tumors and lead to a high rate of cures in both subcutaneous and intra-cranial tumor models. Heteroclitic Epitope Activated Therapy (HEAT) dispenses with the need to identify patient specific neoepitopes and tumor reactive T cells ex vivo. Thus, actively driving a high mutational load in tumor cell vaccines increases their immunogenicity to drive anti-tumor therapy in combination with immune checkpoint blockade.

[1] Department of Molecular Medicine, Mayo Clinic, Rochester, MN 55905, USA. [2] Virology and Gene Therapy Track, Mayo Clinic Graduate School of Biomedical Sciences, Mayo Clinic, Rochester, MN 55905, USA. [3] Department of Immunology, Mayo Clinic, Rochester, MN 55905, USA. [4] Medical Scientist Training Program, Mayo Clinic, Rochester, MN 55905, USA. [5] Department of Biochemistry, Molecular Biology & Biophysics, University of Minnesota, Minneapolis, MN 55455, USA. [6] Center for Cancer and Blood Disorders, Phoenix Children's, Phoenix, AZ 85016, USA. [7] Leeds Institute of Cancer and Pathology (LICAP), Faculty of Medicine and Health, St James' University Hospital, University of Leeds, West Yorkshire, UK. [8] Targeted Therapy Team, Division of Radiotherapy and Imaging, The Institute of Cancer Research, London SW3 6JB, UK. [9] Postgraduate Medical School, University of Surrey, Guildford GU2 7XH, UK. [10] Translational Immunotherapy Team, Division of Radiotherapy and Imaging, The Institute of Cancer Research, London SW3 6JB, UK. [11] Department of Ophthalmology, Mayo Clinic, Rochester, MN 55905, USA. [12] Leeds Cancer Research UK Clinical Centre, Faculty of Medicine and Health, St James' University Hospital, University of Leeds, West Yorkshire, UK. [13]These authors contributed equally: Christopher B. Driscoll, Matthew R. Schuelke. [14]These authors jointly supervised this work: Laura Evgin, Richard G. Vile. *email: vile.richard@mayo.edu

Mutational plasticity of tumors drives malignancy and escape from therapy. In addition to exogenous sources of DNA damage such as radiotherapy and chemotherapeutics[1,2], the APOBEC3 proteins have been shown to be endogenous drivers of genetic heterogeneity in part through their activity as cytidine deaminases[3–7]. In humans there are seven APOBEC3 family members (APOBEC3 A, B, C, D, F, G, and H), while in mice there is one APOBEC3 gene[8]. APOBEC proteins primarily function as host defenses against pathogens and are tightly regulated under normal conditions[9–11]. In particular, APOBEC3B expression is upregulated in multiple cancer types, is associated with the burden of signature C to T mutations, and with poor prognosis and therapeutic resistance[9–14]. We, and others, have shown that high APOBEC3B expression increases escape from oncolytic virotherapy, immunotherapy, and chemotherapy[15,16]. Generally, therefore, high mutational plasticity in tumor cells is regarded as deleterious and many strategies aim to limit ongoing cancer mutation[17].

However, it is also now clear that high mutational loads within tumor cells correlate well with response to cancer immunotherapy, especially immune checkpoint blockade (ICB)[18–20]. Thus, mutation of either passenger or cancer-driving genes generates novel neoepitopes presented by tumor cells. These neoepitopes may be sufficiently different from the unmutated, self-epitopes to prime, and activate, pre-existing T cells which have not been tolerized, or deleted, and which can lead to potentially clearative antitumor T-cell responses[21–24]. Neoepitopes are encoded by nonsynonymous mutations, splice variants, or genome rearrangements that transform a self-peptide into a nonself peptide, and their identification, and approaches to specifically activate their cognate T cells, have provided effective clinical therapy[25]. However, the vast majority of neoepitopes are patient specific, with very few shared neoantigens expressed between patients. Therefore, T cells against each neoepitope must be raised on a patient-by-patient basis, usually from a screen of multiple mutations within the patient's tumor cells, which is time, labor, and cost intensive, thereby limiting their broad applicability.

Exploitation of mutation-induced neoepitopes raises the somewhat counterintuitive possibility that actively driving mutation within cancer cells, rather than inhibiting it, may lead to tumor clearance by sensitizing tumors to immunotherapies. While other groups have explored this idea in the context of inactivation of DNA repair[24], direct comparisons between the deleterious effects of inducing treatment-resistance through mutation, compared to the beneficial consequences of immune-enhancing mutational activity, have not been widely performed. Since APOBEC3B is a driver of cancer genomic diversity, we hypothesize that its over-expression may have two therapeutically opposing consequences. In the first, actively driving mutation of the cellular transcriptome will generate clones which are more aggressive and can evade therapy. Conversely, APOBEC3B-mediated mutation of the cellular immunopeptidome might generate neoepitopes which will prime T-cell responses against the newly immunogenic tumor cells. This hypothesis is consistent with data showing that APOBEC3B expression correlates with increased frequency of tumor infiltrating lymphocytes[3–5,16,26]. In addition, a subset of these novel neoepitopes may represent heteroclitic peptides which prime T cells that recognize both the neoepitope and the unmutated epitope from which it was derived[27,28], consistent with data showing that xenogeneic peptides can generate strong immune responses against native antigens[27–32]. Whereas the priming of T cells reactive only against de novo neoepitopes will allow immune clearance only of APOBEC3B-modified tumor cells themselves, priming of T cells by heteroclitic neoepitopes will allow for bystander clearance of both APOBEC3B-modified, as well as parental, unmodified tumor cells.

We show here that overexpression of APOBEC3B in tumors both increases their ability to evade chemotherapy but also simultaneously confers significantly heightened sensitivity to immunotherapy with ICB, such that tumor cures are achieved in an otherwise very poorly immunogenic murine melanoma tumor model. We also show that APOBEC3B-mediated mutations reproducibly generate novel heteroclitic neoepitopes which activate de novo T-cell responses reactive against the parental, unmodified tumors which are significantly enhanced in their potency by ICB. Finally, we demonstrate that human tumors can be modified through APOBEC3B expression, leading to enhanced T-cell recognition, thereby opening the path to clinical translation but only in a tumor cell vaccine setting, in which the chances of inducing any negative effects through APOBEC3B-mediated induction of increased malignancy would be abrogated by use of irradiated cell vaccines. Thus, the tumor-promoting effects of driving mutation can be uncoupled from its immunogenicity-enhancing effects through the use of heteroclitic epitope activated therapy (HEAT). HEAT is broadly applicable to the treatment of cancers of different histological types and locations and does not require identification of specific neoepitopes on a patient-by-patient basis.

## Results

**APOBEC3B expression and ICB promotes antitumor responses.** We tested the hypothesis that APOBEC3B overexpression drives mutations with the potential to introduce neoepitopes which could be recognized by cognate T cells. B16 melanomas stably expressing the HSVtk (Herpes Simplex Thymidine kinase) suicide gene and either wild-type, active APOBEC3B (APOBEC3B$^{ACTIVE}$) or the catalytically inactive APOBEC3B mutant (APOBEC3B$^{INACTIVE}$)[33] were established by subcutaneous injection of the respective tumor cells and allowed to establish for 7 days. Tumor bearing mice were then treated with ganciclovir (GCV) prodrug therapy, followed by anti-CTLA4 checkpoint inhibitor therapy (Fig. 1a). Although GCV extended survival over untreated controls (Fig. 1b, c) ($p = 0.001$ Log-Rank test with Holm–Bonferroni correction for multiple comparisons), APOBEC3B$^{ACTIVE}$ tumors escaped therapy more quickly than APOBEC3B$^{INACTIVE}$ tumors (Fig. 1b, c) ($p = 0.286$ Log-Rank test with Holm–Bonferroni correction for multiple comparisons). Sequencing of the *HSVtk* gene in the escaped tumors revealed a consistent C-to-T APOBEC3B-signature mutation at base 21, resulting in a premature stop codon. Anti-CTLA4 therapy extended the median survival of mice bearing GCV-treated APOBEC3B$^{INACTIVE}$ tumors (Fig. 1b, left inset), confirming that HSVtk-mediated cell killing is immunogenic[34]. However, anti-CTLA4 transformed the less-effective GCV therapy for APOBEC3B$^{ACTIVE}$ tumors into a sustained, curative treatment (Fig. 1b, right inset) ($p = 0.0045$ Log-Rank test with Holm–Bonferroni correction for multiple comparisons). These data show that while APOBEC3B overexpression drives tumor escape from targeted small molecule therapy, it also significantly enhances recognition and rejection by the immune system in the context of ICB.

**APOBEC3B overexpression and ICB improves vaccine efficacy.** To exploit the mutational burden generated by APOBEC3B expression without contributing to tumor evolution, we generated an APOBEC3B-modified tumor vaccine. Mice bearing subcutaneous B16 parental tumors were treated with B16 cell lysates retrovirally transduced to express either APOBEC3B$^{ACTIVE}$ or

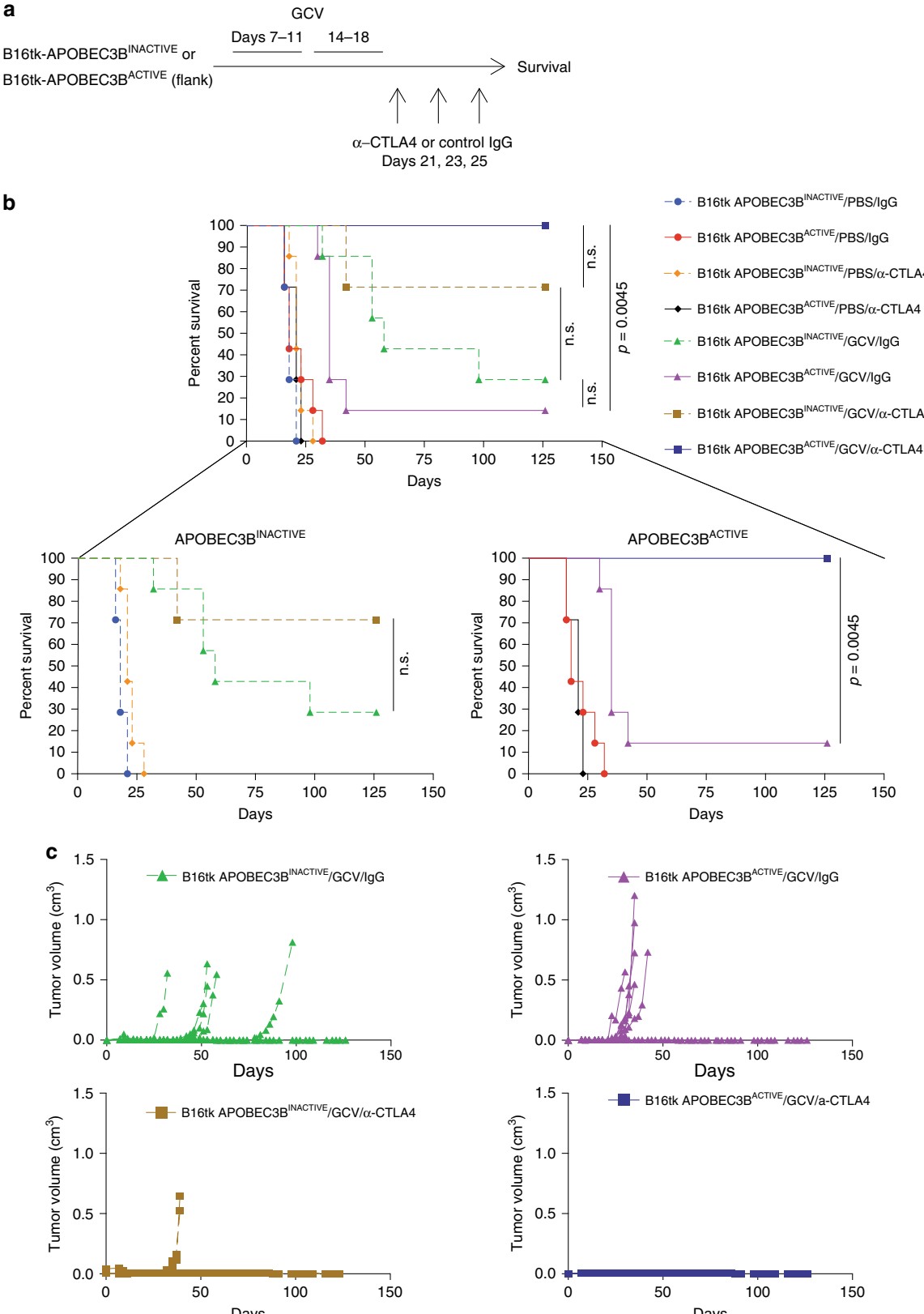

**Fig. 1 APOBEC3B expression enhances tumor escape but sensitizes tumors to immune checkpoint blockade therapy. a** On day 0, $2 \times 10^5$ B16TK cells expressing either APOBEC3B^INACTIVE or APOBEC3B^ACTIVE were subcutaneously implanted into the right flank of C57Bl/6 mice. Two 5-day courses of GCV therapy (50 mg/kg i.p.) were administered from days 7 to 11, and 14 to 18, followed by anti-CTLA4 antibody or control IgG (5 mg/kg i.p.) on days 21, 23, and 25. **b, c** Mice were treated according to the regimen in (**a**) (n = 7 mice/group) tumor volumes were measured three times per week (**c**) and sacrificed when tumors grew above 1 cm in length or width. Representative of three separate experiments. Groups were compared using a Log-Rank test with Holm–Bonferroni correction for multiple comparisons.

APOBEC3B$^{INACTIVE}$, followed by anti-PD1 or control immunoglobulin (IgG) (Fig. 2a). The APOBEC3B$^{ACTIVE}$ vaccine alone significantly delayed tumor growth (Fig. 2b) and prolonged survival (Fig. 2c) compared to the B16-APOBEC3B$^{INACTIVE}$ vaccine ($p = 0.0005$ Log-Rank test with Holm–Bonferroni correction for multiple comparisons). However, the addition of anti-PD1, anti-CTLA-4, or both ICB therapies induced complete and sustained tumor regressions in mice treated with the B16-APOBEC3B$^{ACTIVE}$ vaccine, but not in mice treated with the B16-APOBEC3B$^{INACTIVE}$ vaccine (Fig. 2c, d) ($p = 0.0008$ Log-Rank test with Holm–Bonferroni correction for multiple comparisons). B16 tumor vaccines administered to mice bearing subcutaneous TC2 syngeneic prostate tumors were ineffective, even with APOBEC3B$^{ACTIVE}$/anti-PD1 therapy (Fig. 2c), suggesting the induced T-cell response reflected the specific immunopeptidome of the vaccine. We did not observe any symptoms of autoimmunity in any of these experiments.

**Vaccination elicits a T and NK cell antitumor response.** Depletion of CD4, CD8, and NK cells significantly reduced the efficacy of B16-APOBEC3B$^{ACTIVE}$/anti-PD1 therapy (Supplementary Fig. 1A) ($p = 0.0004$, 0.0008, and 0.026, respectively Log-Rank test with Holm–Bonferroni correction for multiple comparisons) and 7 of 7 mice which rejected their primary tumors in Supplementary Fig. 1 following treatment with B16-APOBEC3B$^{ACTIVE}$ + α-PD1 also rejected a further re-challenge with $2 \times 10^5$ parental B16 cells on the opposite flank at day 150 post primary challenge, indicating the induction of a memory T-cell response.

When cocultured in vitro with parental, unmodified tumor cells, splenocytes from mice which received B16-APOBEC3B$^{ACTIVE}$ vaccines secreted significantly more interferon gamma (IFNγ) than mice treated with B16-APOBEC3B$^{INACTIVE}$ (Supplementary Fig. 1B) ($p < 0.05$). The in vitro recall response was significantly increased when B16-APOBEC3B$^{ACTIVE}$ vaccination was combined with ICB in vivo. Adoptive transfer of CD8 T cells from mice treated with the B16-APOBEC3B$^{ACTIVE}$/anti-PD1 vaccine, but not the B16-APOBEC3B$^{INACTIVE}$/anti-PD1 vaccine, conferred partial protection against parental B16 tumors (Supplementary Fig. 1C) ($p = 0.0009$ Log-Rank test with Holm–Bonferroni correction for multiple comparisons).

**APOBEC3B vaccine is therapeutic in a brainstem glioma model.** Mice bearing brainstem-implanted GL261 tumors were treated with GL261-APOBEC3B$^{ACTIVE}$ or GL261-APOBEC3B$^{INACTIVE}$-modified GL261 vaccines (Fig. 3a). Mice treated with the GL261-APOBEC3B$^{ACTIVE}$-modified vaccine survived significantly longer than untreated controls (Fig. 3b) ($p = 0.0015$ Log-Rank test with Holm–Bonferroni correction for multiple comparisons). The GL261-APOBEC3B$^{ACTIVE}$/anti-PD1 vaccine doubled median survival, although when controlling for multiple comparisons this did not reach statistical significance compared to untreated controls ($p = 0.076$ Log-Rank test with Holm–Bonferroni correction for multiple comparisons). However, the combination of anti-CTLA4 and anti-PD1 therapy with the GL261-APOBEC3B$^{ACTIVE}$ vaccine significantly improved survival compared to mice treated with control immunoglobulin, with all mice surviving past 40 days ($p = 0.0012$ Log-Rank test with Holm–Bonferroni correction for multiple comparisons). Splenocytes from mice treated with the GL261-APOBEC3B$^{ACTIVE}$ vaccine with ICB secreted significantly increased levels of IFNγ on restimulation with parental GL261 tumors compared to mice receiving only the GL261-APOBEC3B$^{ACTIVE}$ vaccine (Fig. 3c) ($p < 0.01$ one-way

ANOVA). These data show that the APOBEC3B$^{ACTIVE}$-modified vaccine and ICB strategy is effective against a second tumor type and retains efficacy in the immunologically distinct site of the brainstem.

**APOBEC3B induces heteroclitic neoepitopes.** We have previously shown that vesicular stomatitis virus (VSV) infection of B16tk-APOBEC3B$^{ACTIVE}$ cells led to enhanced escape from oncolytic VSV therapy compared to B16tk-APOBEC3B$^{INACTIVE}$ cells, highlighting the role of APOBEC3B in evading primary therapies[16]. A whole-genome sequencing screen of the B16tk-APOBEC3B$^{ACTIVE}$-modified, VSV-escaped population, compared to B16tk parental cells, identified over 1,000,000 mutations or indels unique to the B16-APOBEC3B$^{ACTIVE}$ overexpressing cells (SRA Submission: SRP159367). Using an in vitro cytidine deamination assay (Supplementary Fig. 2A), we confirmed that, whereas the APOBEC3B$^{ACTIVE}$ vector induced cytidine deaminase activity consistent with APOBEC mutagenesis, the APOBEC3B$^{INACTIVE}$ vector did not. Consistent with these data, whilst we detected APOBEC3B-signature mutations in the B16-APOBEC3B$^{ACTIVE}$ cells, these were not, or very rarely, present in the corresponding B16-APOBEC3B$^{INACTIVE}$ cells (Fig. 4b). However, in addition to core, APOBEC3B-signature mutations, multiple accessory mutations were also generated in the APOBEC$^{ACTIVE}$ cells compared to the APOBEC$^{INACTIVE}$ cells, including nucleotide changes and indels which occurred typically at low clonalities in the vaccine cells (Supplementary Fig. 2B). Of the ~1,000,000 total mutations detected by NGS in the B16-APOBEC3B$^{ACTIVE}$ cells, about 68,000 contained the classical APOBEC mutational signature. Of these ~68,000 mutations, 244 resulted in predicted amino acid changes in expressed proteins through C to T or G to A transitions leading to missense mutations consistent with APOBEC3B mutational activity (Supplementary Data 1)[35]. These mutations were converted into 21-amino acid sequences, with 10 amino acids flanking the mutational site (Supplementary Data 2). Totally, 8–10 mer peptides were generated from these sequences, placing the mutated amino acid in every position from the amino terminal to the carboxy terminal end. These peptides were then filtered through an MHC binding affinity algorithm (Supplementary Table 1). Because our depletion studies showed that CD8 T cells were critical for therapy, we hypothesized that a proportion of APOBEC3B-induced neoepitopes in the vaccine cells would also act as heteroclitic peptides and screened the potential neoepitopes for differential binding to MHC Class I[29]. Thresholds were set for binding affinity to either H2K$^b$ (Fig. 4a, dark gray and dark blue) or H2D$^b$ (Fig. 4a, light gray and light blue), such that wild-type peptide binding was above 500 nM, while APOBEC3B-mutated peptide binding was below 500 nM[36,37]. By refining this list using the EMBL-EBI Expression Atlas, eight candidate proteins were identified that are expressed in the skin: Ahctfl, C77080, Csde1, Fcgbp, Plbd2, Smc4, Stag2, and Xpo1 (which contained three candidate peptide sequence variants) (Fig. 4a). Of these APOBEC-characteristic mutations, none were present clonally in the B16-APOBEC$^{ACTIVE}$ vaccine population (Fig. 4b) and varied in frequency between ~60 and <10%. Only one of the mutations (Stag2) could be detected by NGS to be present in the ABPOC3B$^{INACTIVE}$ cells, whereas all of the others were induced by de novo APOBEC3B activity.

**In vitro validation of APOBEC3B-induced neoepitopes.** B16 cell lines were generated that transiently overexpressed the ten wild-type (B16-WT sequence) peptides or the mutant potential neoepitope peptides (B16-APOBEC3B-modified sequence). Expression constructs contained the ten wild type, or neoepitope,

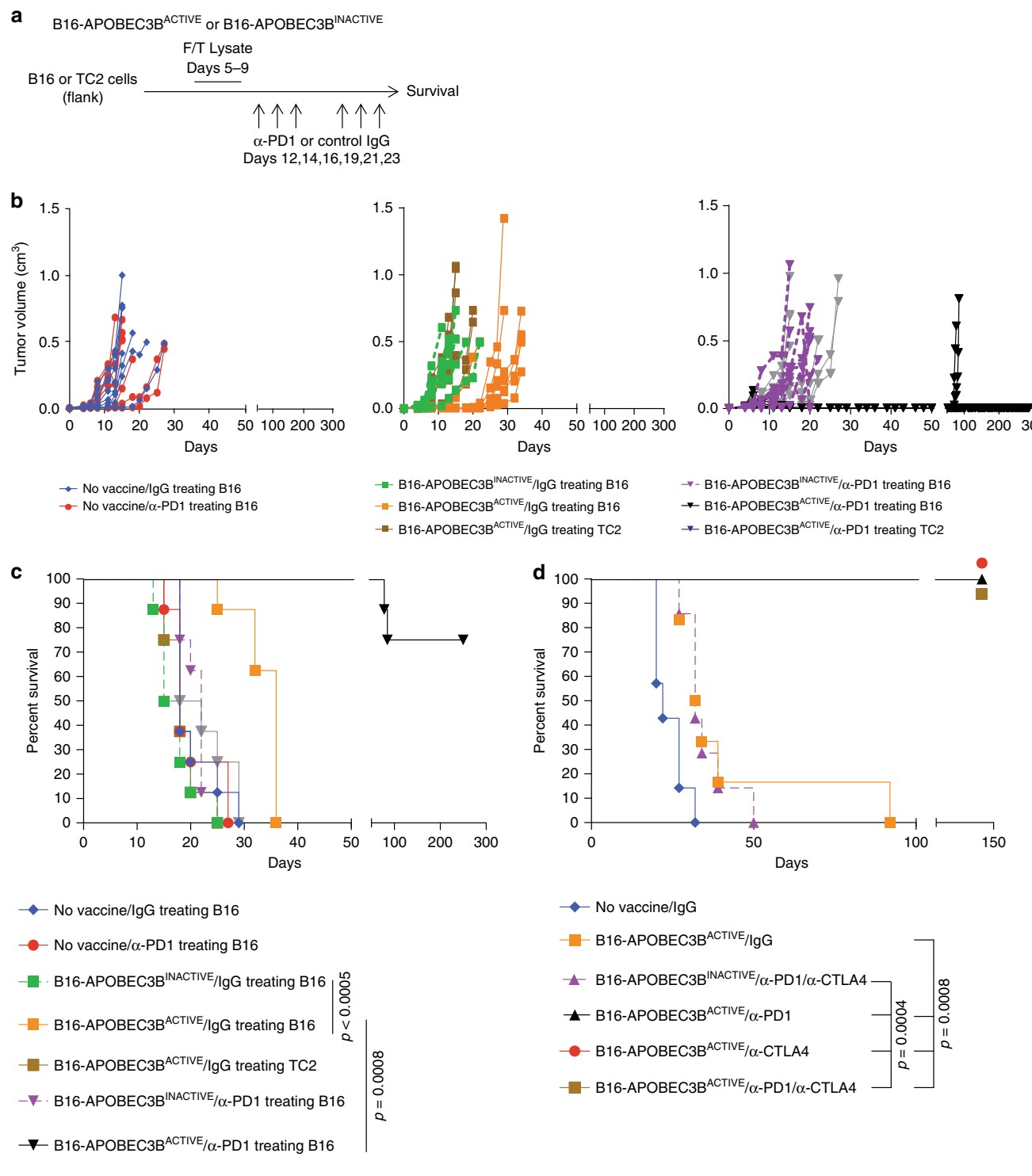

**Fig. 2 APOBEC3B enhances B16 cell lysate vaccine and synergizes with immune checkpoint blockade therapy. a** On day 0, $2 \times 10^5$ B16 murine melanoma or TC2 murine prostate carcinoma were implanted subcutaneously into the right flank of C57Bl/6 mice. One 5-day course of APOBEC3B$^{ACTIVE}$ or APOBEC3B$^{INACTIVE}$-modified B16 cell vaccines (freeze/thaw lysate of $10^6$ cells i.p.) was administered from days 5 to 9. This was followed by anti-PD1 antibody or IgG control (12.5 mg/kg i.p.) on days 12, 14, 16, 19, 21, and 23. **b** Mice ($n = 8$/group) were treated according to the regimen in (**a**) and tumor volumes were measured three times per week. **c** Kaplan–Meier survival curves representing experiment described by (**a**) and (**b**). Representative of 5 experiments. **d** Treatment of subcutaneous B16 murine melanomas as in (**a**), except with the addition of anti-CTLA4 antibody (5 mg/kg i.p.) as a monotherapy or in combination with anti-PD1 ($n = 7$ mice/group). The black triangle, red circle, and brown square symbols represent three separate groups of 7 mice treated with B16-APOBEC3B$^{ACTIVE}$ vaccine with anti-PD1, anti-CTLA4, or anti-PD1/anti-CTLA4 antibody therapy that all had 100% survival at 150 days. This experiment was repeated once. Groups were compared using a Log-Rank test with Holm–Bonferroni correction for multiple comparisons.

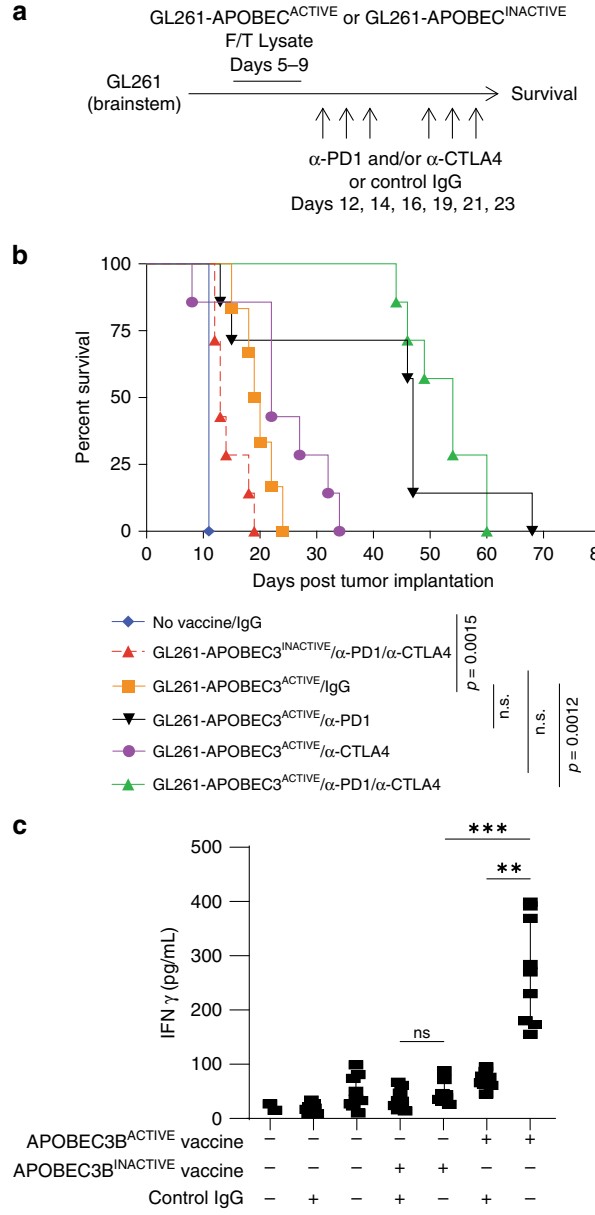

**Fig. 3 APOBEC3B-modified GL261 cell lysate vaccine also effective against orthotopic gliomas. a** Treatment strategy from (Fig. 2a) was adapted to target GL261 murine gliomas. In summary, on day 0, $5 \times 10^4$ GL261 murine glioma cells were implanted into the brainstem of C57Bl/6 mice. One 5-day course of APOBEC3B$^{ACTIVE}$ or APOBEC3B$^{INACTIVE}$-modified GL261 cell vaccines (freeze/thaw lysate of $10^6$ cells i.p.) was administered from days 5 to 9. This was followed by anti-PD1 antibody, anti-CTLA4 antibody or IgG control (12.5 mg/kg i.p.) on days 12, 14, 16, 19, 21, and 23. **b** Kaplan–Meier survival curves representing experiment described by (**a**) ($n = 7$ mice/group). Representative of three separate experiments. Groups were compared using a Log-Rank test with Holm–Bonferroni correction for multiple comparisons. **c** Spleens and lymph nodes obtained from mice treated with GL261-APOBEC3B$^{ACTIVE}$ vaccination and/or antibody-mediated checkpoint inhibition were made into single-cell suspensions and cocultured with GL261 target cells for 72 h. Supernatant from the coculture was assayed using a mouse interferon gamma ELISA. ANOVA was used followed by Tukey's multiple comparison test. Data points represent three separate biologic replicates. Error bars indicate mean and SD. **$p \leq 0.01$, ***$p \leq 0.001$.

peptides separated by Ala-Ala-Tyr (AAY) spacers and tagged with an N-end degron motif to increase MHC Class I presentation (Fig. 4c)[38,39]. Splenocytes from unvaccinated or B16-APOBEC3B$^{INACTIVE}$ vaccine-treated mice showed no IFNγ recall response to cells overexpressing B16-WT peptides or B16-APOBEC3B-modified peptides (Fig. 4d). Splenocytes from B16-APOBEC3B$^{ACTIVE}$ vaccine-treated mice—but without anti-PD1 ICB—secreted very low levels of IFN-γ in response to restimulation with B16 cells expressing only wild-type epitopes (Fig. 4d) ($p < 0.0001$ two-way ANOVA). In contrast, splenocytes from B16-APOBEC3B$^{ACTIVE}$ vaccine-treated mice secreted significantly higher levels of IFN-γ in response to restimulation with B16 cells expressing the string of APOBEC3B-modified epitopes described in Fig. 4c. These data suggest that APOBEC3B modification of B16 cells induces mutations which stimulate T-cell responses against at least one of the epitopes included in the APOBEC3B string.

Therefore, expression constructs were created for each of the individual neoepitopes. As before (Fig. 4d), splenocytes from naïve mice, or from mice vaccinated with the B16-APOBEC3B$^{INACTIVE}$ vaccine, did not generate a recall response against parental B16 tumor cells in vitro (Fig. 4e). However, splenocytes from B16-APOBEC3B$^{ACTIVE}$ vaccine-treated mice—and in the presence of anti-PD-1 ICB—secreted significant levels of IFN-γ in response to restimulation with parental B16 cells, or parental B16 cells expressing only wild-type epitopes (Fig. 4e). These data show that the addition of anti-PD1 ICB enhances the recognition of self, unaltered epitopes by T cells raised against APOBEC3B-mutated neo-epitopes. Moreover, these splenocytes also secreted significantly higher levels of IFN-γ in response to restimulation with B16 cells expressing the string of APOBEC3B-modfied epitopes (consistent with Fig. 4d in the absence of ICB). This increased recognition of B16 cells by the expression of the string of APOBEC3B-modified epitopes was only reproduced by single epitope expression of the APOBEC3B-modified cold shock domain-containing E1(CSDE1*) epitope but not the wild-type CSDE1 epitope (Fig. 4d). Overexpression of none of the remaining nine potential predicted APOBEC3B-modified neo-pitopes enhanced the recall response above that seen in response to parental B16 cells themselves (Fig. 4e). These data confirm that treatment with the B16-APOBEC3B$^{ACTIVE}$ vaccine and anti-PD1 induced T-cell responses against parental B16 tumors and suggest that an APOBEC3B-mutated neoepitope within CSDE1 may function as a heteroclitic neoepitope.

To validate that the putative CSDE1 heteroclitic neoepitope was indeed present in the vaccine preparation, we sequenced the CSDE1 gene in B16-APOBEC3B$^{ACTIVE}$ vaccine cells used in Figs. 2 and 3. Consistent with the lack of APOBEC3B deaminase activity of the APOBEC3B$^{INACTIVE}$ construct (Supplementary Fig. 2A), B16 parental and B16-APOBEC3B$^{INACTIVE}$ cell vaccines contained only the wild-type ATGAGCTTTGATCCA sequence (Fig. 5a, b). However, the vaccine preparation contained a mixed population of cells carrying either the wild-type ATGAGCTTTGATCCA sequence, as found homogeneously in the parental B16 and B16-APOBEC3B$^{INACTIVE}$ vaccine populations, or the mutated ATGAGCTTTGATTCA sequence (Fig. 5c), which encodes the potentially heteroclitic CSDE1 neoepitope (Fig. 4e). We further validated that the CSDE1 mutation is a reproducible and consistent target of APOBEC3B activity in B16 cells in two additional vaccine preparations (Supplementary Fig. 3).

**The CSDE1 heteroclitic neoepitope primes anti-B16 T cells.** B16 cells transfected with the APOBEC3B-mutated CSDE1 epitope (CSDE1*), or the wild-type parental epitope (CSDE1), were

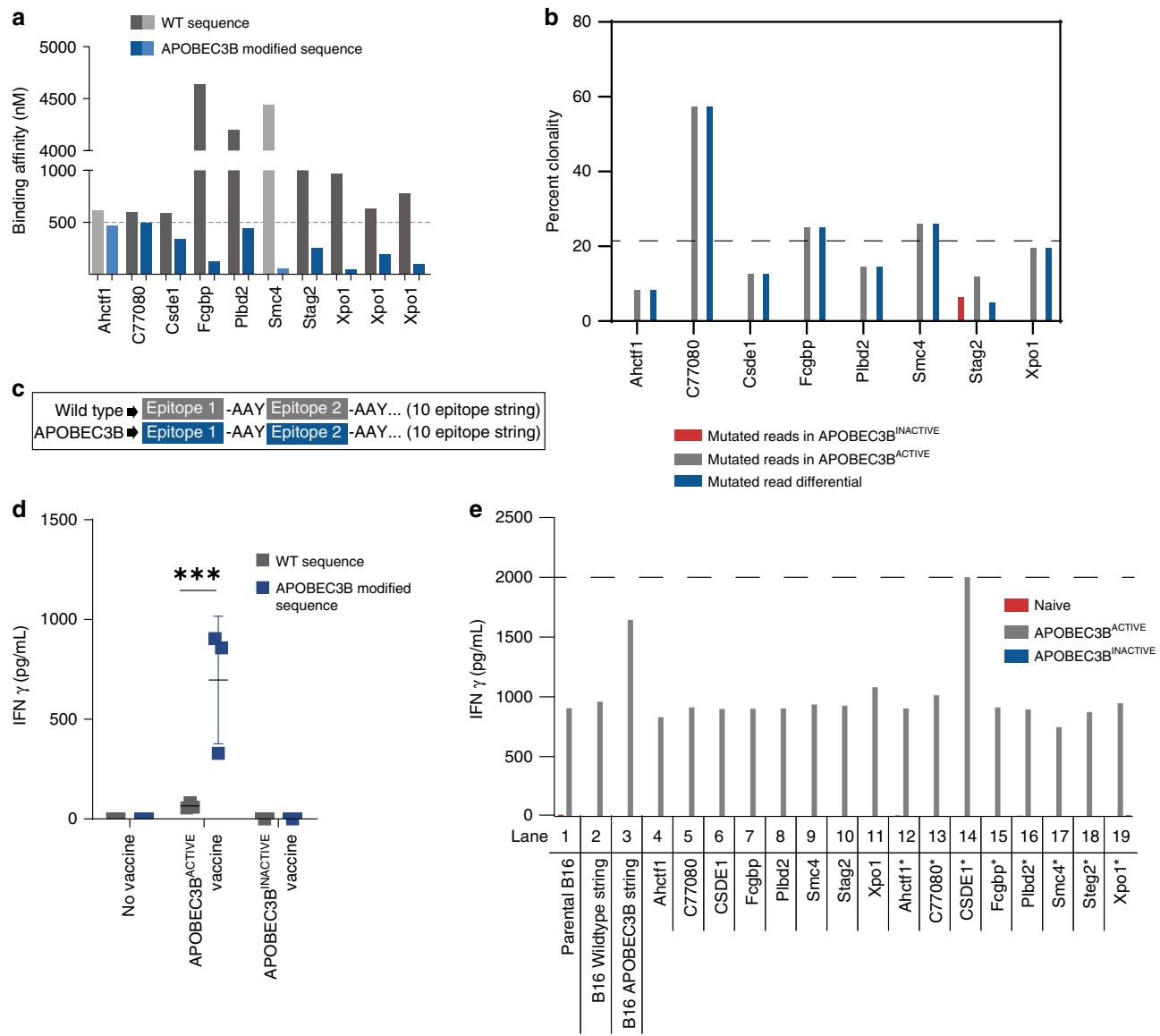

**Fig. 4 Mutated peptides with differential MHC binding were identified as potential heteroclitic neoepitopes. a** The seven genes leading to ten peptides that showed differential MHC I binding between the wild-type and mutated peptide are shown. The threshold for binding affinity was set at 500 nM with weak binding being above 500 nM and strong binding being below 500 nM. **b** The clonalities of the genes in (**a**) is shown for both the B16-APOBEC3B[ACTIVE] and B16-APOBEC3B[INACTIVE] cell populations. **c** Expression plasmids were constructed that encoded either all ten wild type (wild-type epitope string) or all ten mutated peptide sequences (APOBEC3B epitope string) with AAY spacers between each peptide to enhance cleavage and presentation. **d** C57Bl/6 mice were left unvaccinated (no vaccine) or were vaccinated with cell lysates of B16-APOBEC3B[ACTIVE], or B16-APOBEC3B[INACTIVE] cells, with no added α-PD1 ICB, as described in Fig. 2a (n = 3 mice per group). Splenocytes from these mice were cocultured with parental B16 tumors that overexpressed either the wild type, or APOBEC3B-modified, epitope string (**c** above) and IFN-γ levels in the supernatants were measured by ELISA. Data points represent three biologic replicates. Error bars indicate mean and SD. A two-way ANOVA was used followed by Tukey's multiple comparison test. ***p ≤ 0.001. **e** C57Bl/6 mice were left unvaccinated (Naïve) or were vaccinated with cell lysates of B16-APOBEC3B[ACTIVE], or B16-APOBEC3B[INACTIVE] cells, with immune checkpoint blockade (αPD-1) as described in Fig. 2a. Splenocytes from these mice were cocultured with (lane 1) parental B16 tumors, or with parental B16 cells expressing (lane 2) wild-type string of epitopes (**c** above), or (lane 3) the APOBEC3B string of epitopes (**c** above), or expressing the individual wild-type epitopes (lanes 4–11), or the individual mutated epitopes (lanes 12–19). IFN-γ levels in the supernatants were measured by ELISA and are shown from splenocytes from the naïve (red), B16-APOBEC3B[ACTIVE] (gray), or B16-APOBEC3B[INACTIVE] vaccinated mice (blue). Dashed line indicates upper limit of detection. Data shown represent single well results from pooled splenocytes from three mice per vaccination group which are representative of multiple experiments.

used as a vaccine in combination with anti-PD1 to treat parental B16 tumors in the flank or brainstem. Mice vaccinated with B16 cells expressing CSDE1 succumbed to disease at the same time as untreated controls (Fig. 5d, e) (p = 0.7805 and 0.158 Log-Rank test with Holm–Bonferroni correction for multiple comparisons). B16 cells expressing CSDE1* generated partial therapy (p = 0.0009 and

0.002 Log-Rank test with Holm–Bonferroni correction for multiple comparisons), whereas the B16-APOBEC3B[ACTIVE] vaccine was significantly more effective, highlighting the value of vaccination with a wide range of antigens to prevent tumor escape (p = 0.0442 and 0.001 Log-Rank test with Holm–Bonferroni correction for multiple comparisons). When splenocytes from these mice

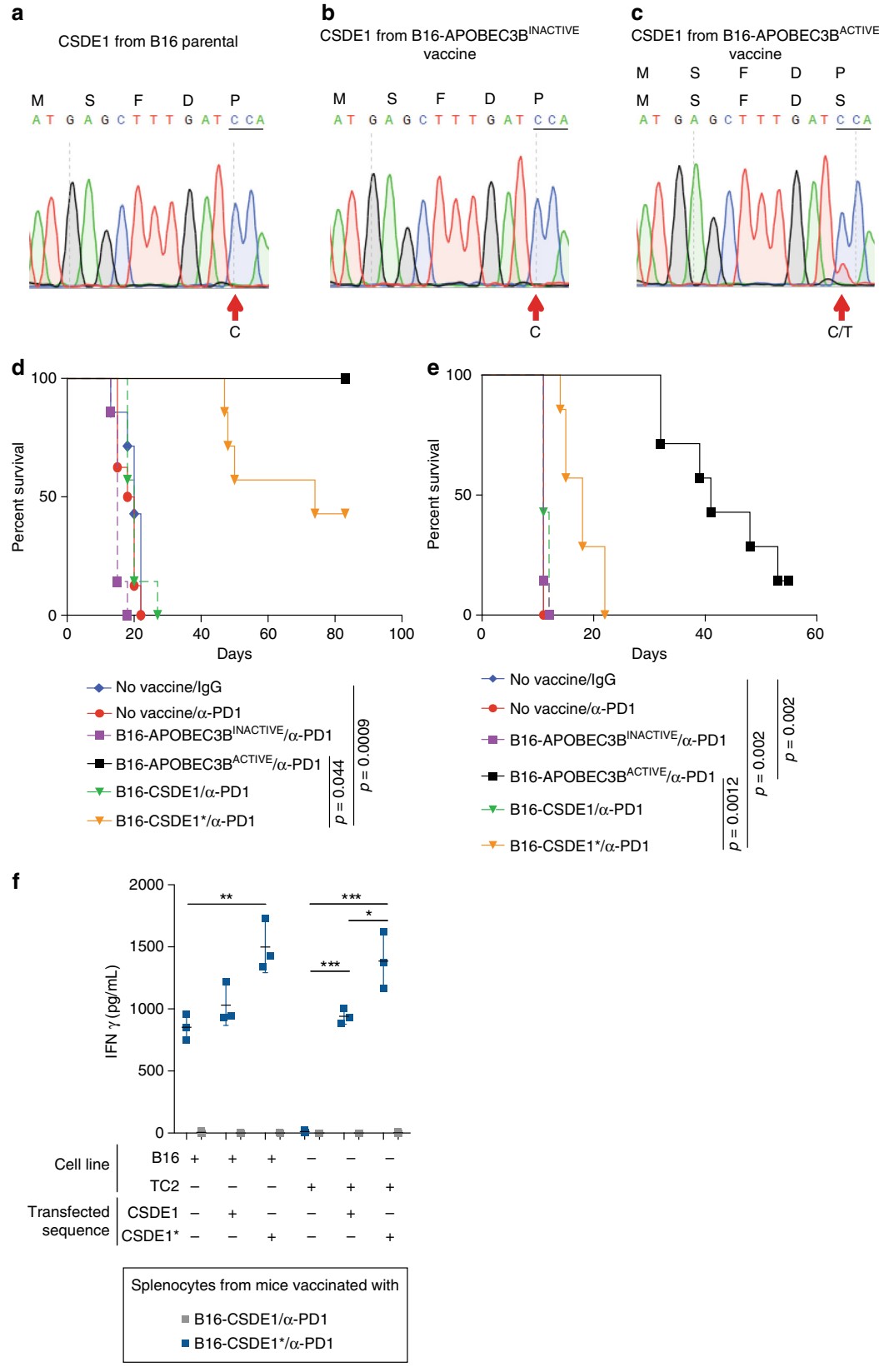

were cocultured with either parental B16 or TC2 prostate tumor cells expressing either CSDE1 or CSDE1* epitopes, only mice vaccinated with B16-CSDE1* and treated with ICB showed an IFNγ recall response against parental B16 cells (Fig. 5f) ($p < 0.01$ one-way ANOVA). Most importantly, in the context of an irrelevant TC2 tumor, mice vaccinated with B16-CSDE1* reacted against both the TC2-expressed CSDE1* and, albeit to a lesser extent, against the CSDE1 epitope, thereby confirming that the APOBEC3B-mutated CSDE1* acts as a heteroclitic peptide ($p < 0.001$ one-way ANOVA).

**Fig. 5 Sequencing of APOBEC3B^ACTIVE modified vaccines generates reproducible mutations in CSDE1.** Sanger sequencing of CSDE1 from **a** parental B16 cells, **b** APOBEC3B^ACTIVE modified vaccine, and **c** APOBEC3B^INACTIVE modified vaccine was performed. Figures are representative of three independent experiments. Each preparation of the APOBEC3B^ACTIVE modified vaccine had proportions of cells containing a C or a T at the 13th base pair, corresponding to the P5S amino acid change seen in Fig. 5 and Supplementary Fig. 2. (Figure was prepared using SnapGene software (from GSL Biotech; available at snapgene.com). **d** On day 0, $2 \times 10^5$ B16 murine melanoma cells were implanted subcutaneously into the right flank of C57Bl/6 mice. Two 5-day courses of B16-APOBEC3B^ACTIVE, B16-APOBEC3B^INACTIVE, B16-CSDE1, or B16-CSDE1* vaccines (freeze/thaw lysate of $1 \times 10^6$ cells i.p.) were administered from days 5 to 9 and 12 to 16. This was followed by anti-PD1 antibody or IgG control (12.5 mg/kg i.p.) on days 12–16, 19, 21, and 23. Kaplan–Meier survival curves representing experiment described. Representative of three separate experiments. **e** On day 0, $5 \times 10^4$ B16 murine melanoma cells were implanted into the brainstem of C57Bl/6 mice. One 5-day course of B16-APOBEC3B^ACTIVE, B16-APOBEC3B^INACTIVE, B16-CSDE1, or B16-CSDE1*-modified cell vaccines (freeze/thaw lysate of $10^6$ cells i.p.) was administered from days 5 to 9. This was followed by anti-PD1 antibody or IgG control (12.5 mg/kg i.p.) on days 12, 14, 16, 19, 21, and 23. Kaplan–Meier survival curves representing experiment described ($n = 7$ mice/ group). This experiment was repeated once. **f** Spleens and lymph nodes obtained from mice treated with B16-CSDE1 or B16-CSDE1* vaccination and antibody-mediated checkpoint inhibition in (**e**) were made into single-cell suspensions and cocultured with B16 or TC2 target cells expressing CSDE1 or CSDE1* for 72 h. Supernatant from the coculture was assayed using a mouse interferon gamma ELISA. Data points represent three biologic replicates. Error bars indicate mean and SD. ANOVA was used followed by Tukey's multiple comparison test. **$p \leq 0.01$, ***$p \leq 0.001$.

**Human reactivity to APOBEC3B-modified tumors.** Our murine data above justifies the clinical translation of APOBEC3B-modification of tumor cells as cancer vaccines. To validate this approach, human melanoma cells, APOBEC3B-modified or controls, were cocultured in vitro with activated human CD3 T cells for 10 days. Following this period of exposure to putative antigens presented by melanoma cells, the T cells were restimulated with unmodified tumors and assayed for interferon-gamma (IFNγ) secretion, proliferation, and target cell killing (Fig. 6a). As a mimic of the clinical scenario in which a patient's own tumor cells would be used as the platform for the APOBEC3B-modified vaccine, Mel888-APOBEC3B^ACTIVE or Mel888-GFP cells were used to educate donor T cells against unmodified Mel888 cells. Consistent with our murine data, while T-cell activation in the presence of Mel888-GFP cells induced IFNγ secretion, proliferation, and tumor killing, Mel888-APOBEC3B^ACTIVE coculture significantly enhanced each of these T-cell functions (Fig. 6b) ($p = 0.00636$, $0.00285$, and $0.00479$, respectively Student's $t$ test). Addition of anti-PD1 checkpoint antibodies further increased T-cell activity both in cells educated by GFP- ($p = 0.00468$ and $p = 0.01996$) and APOBEC3B^ACTIVE-modified Mel888 tumors ($p = 0.49602$ and $p = 0.0233$ Student's $t$ test).

Finally, and also consistent with our murine data, we confirmed that the use of APOBEC3B modification to stimulate human antitumor T-cell responses was applicable across tumor types. In vitro activated T cells were cocultured with autologous, monocyte-derived dendritic cells pulsed with lysates from either a GFP-modified human pediatric glioma cell line (SJPDGF1), with lysate from the APOBEC3B^ACTIVE-modified SJPDGF1 equivalent, or with lysate of APOBEC3B^ACTIVE-modified Mel888 cells. These in vitro primed T cells were then restimulated with dendritic cells pulsed with unmodified SJPDGF1 lysate (Fig. 6c). T cells primed with DC pulsed with SJPDGF1-APOBEC3B^ACTIVE-modified lysate secreted increased levels of IFNγ compared to irrelevant (Mel888) or unmodified (SJPDGF1) tumor lysates, which further increased in the presence of anti-PD1 antibodies (Fig. 6d). Together, these data show that APOBEC3B-modification enhances the immunogenicity of human tumor cells for T-cell recognition and that such an approach can potentially be translated clinically in the context of an autologous vaccine platform.

## Discussion

Cancer evolution is driven by somatic mutation, leading to selection of subpopulations which become progressively more malignant and which can evade therapy[40,41]. Generally, therefore, high mutational plasticity is regarded as deleterious. However, mutational plasticity also generates neoepitopes, which can prime antitumor T-cell responses[21–24]. Therefore, we tested here the controversial concept that instead of preventing ongoing mutation in tumor cells, actively driving a high mutational load through APOBEC3B expression may enhance therapy by rendering poorly immunogenic tumors susceptible to immunotherapy.

As expected, B16tk tumors expressing APOBEC3B (APO-BEC3B^ACTIVE) were more resistant to frontline GCV chemotherapy than control tumors, consistent with the mutagenic activity of APOBEC3B selecting clones insensitive to GCV. However, that same APOBEC3B mutagenesis also conferred significantly heightened sensitivity to immunotherapy with ICB. We hypothesize that APOBEC3B-induced mutations generated neoepitopes which became potent targets for de novo T-cell responses and that GCV-mediated killing enhanced the ability of antigen presenting cells to present these neoepitopes to potentially reactive T cells. Thus, loss of frontline therapy due to mutation-driven resistance may be more than compensated for by therapeutic gain from enhanced mutation-driven immunogenicity. These data open the way for new therapeutic approaches based upon turning poorly immunogenic (cold) tumors into highly immunogenic and well infiltrated (hot) tumors by actively driving mutation to corrupt the cellular immunopeptidome and express neoepitopes de novo.

However, this approach also risks simultaneously driving treatment resistance and increased malignancy by mutation to cancer driver genes. Therefore, we exploited the immunogenicity of APOBEC3B-induced heteroclitic neoepitopes in the setting of a therapeutic vaccine. In this model, T cells which escape central tolerance and may be weakly reactive to self tumor antigens would be activated by a heteroclitic neoepitope (induced by APOBEC3B mutation) and react back against a host tumor expressing the native protein[27–32,42]. APOBEC3B expression would generate a library of mutated antigens, raising multiple clones of tumor-reactive T cells, thereby reducing the chances of antigen escape. Consistent with this hypothesis, the modest survival enhancement generated by the B16-APOBEC3B^ACTIVE vaccine alone was converted into a curative treatment when combined with ICB, dependent upon CD4, CD8, and NK cells. Interestingly, even in the presence of ICB to overcome immunosuppressive mechanisms acting on the activated antitumor T-cell response[43–46], we did not observe any signs of autoimmunity. However, in each type of tumor which might be treated by this approach, detailed preclinical histological and toxicological studies should be performed to confirm that heteroclitic T-cell responses against APOBEC3B-mutated neo-epitopes do not raise autoreactive T-cell activity against critically important self-peptide epitopes expressed on the corresponding normal tissues.

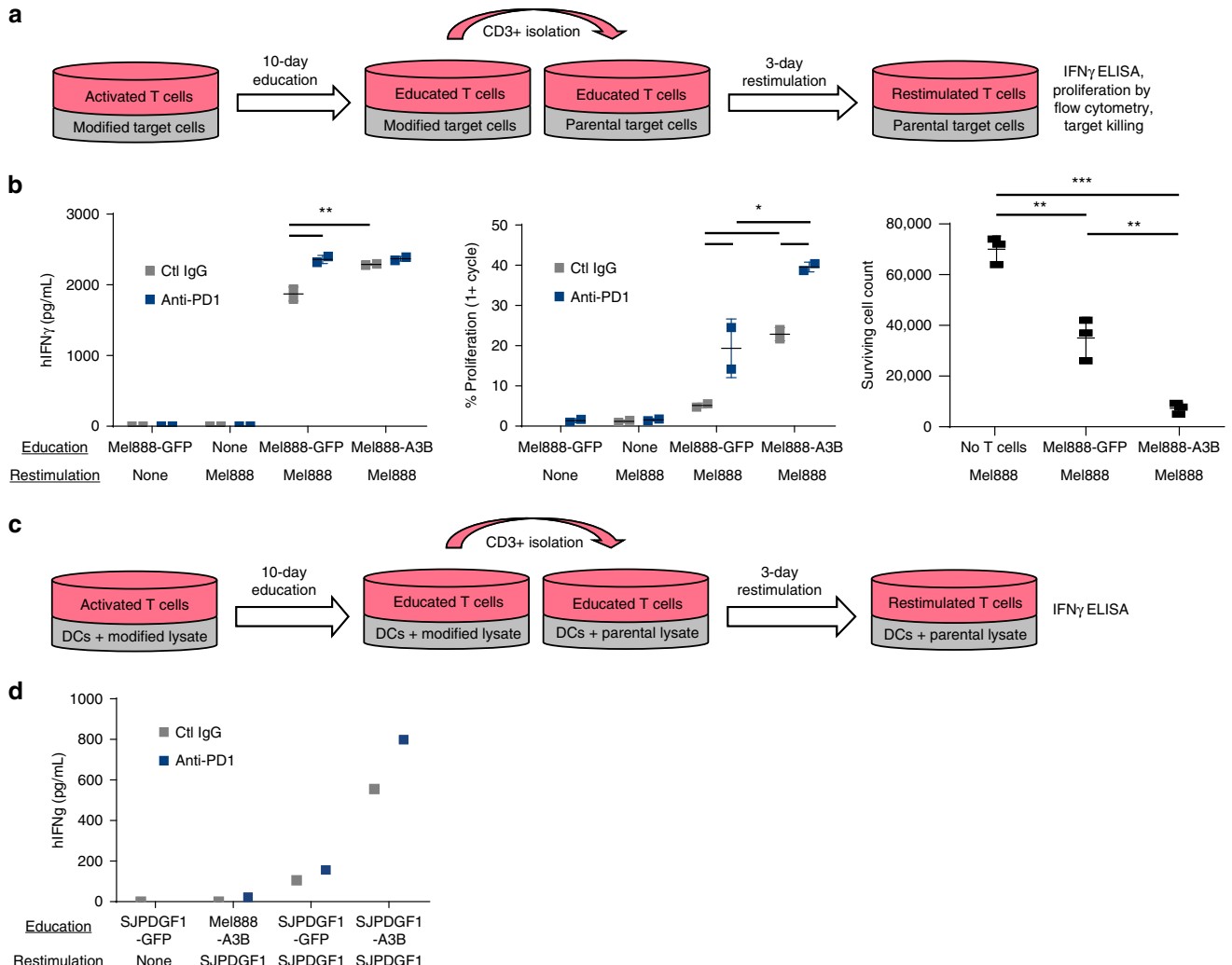

**Fig. 6 Human reactivity to APOBEC3B-modified tumors. a** CD3+ T cells from healthy donor PBMCs were isolated and activated with CD3/CD28 beads. These T cells were cocultured with Mel888 cells previously transduced by lentivirus expressing GFP or APOBEC3B and pretreated for 12 h with human interferon gamma (hIFNγ). After 10 days of co-incubation, CD3+ T cells were isolated, stained with cell trace violet, and replated with hIFNγ pretreated Mel888 parental cells. After 3 days, supernatant was collected for hIFNγ ELISA, T cells underwent flow cytometric analysis for proliferation by cell trace violet dilution, and Mel888 cells were counted to assess target killing. Representative of three separate experiments. **b** hIFNγ ELISA, T-cell proliferation, and target killing from T cells cocultured with autologous Mel888 cells for both education and restimulation. Error bars indicate mean and SD. **c** Prior to coculture, CD14+ cells were isolated from healthy donor PBMCs and matured into monocyte-derived dendritic cells. CD3+ T cells were isolated from the same donor PBMCs and activated with CD3/CD28 beads. These T cells were cocultured with the mature dendritic cells and pulsed with pediatric glioma (SJPDGF1) or Mel888 lysate previously transduced by lentivirus expressing GFP or APOBEC3B. Lysates were added again on days 2 and 3 of coculture. Seven days later, CD3+ T cells were isolated, and cocultured with fresh monocyte-derived dendritic cells pulsed with parental SJPDGF1 lysate. Three days later, supernatant was collected for hIFNγ ELISA. **d** hIFNγ ELISA from vaccination using Mel888 or SJPDGF1 lysate for education and SJPDGF1 lysate for restimulation. All cocultures were performed at a 10:1 ratio of targets:effectors. Student's *t* test was performed followed by Holm–Bonferroni correction for multiple comparisons. *$p \leq 0.05$, **$p \leq 0.01$, ***$p \leq 0.001$.

Our broadly mutated vaccine was also effective against gliomas in the immunologically distinct location of the brainstem[47–49]. The fact that anti-CTLA-4 improved vaccine efficacy in the B16 flank tumor model, but not in the GL261 intracranial model, is consistent with ICB therapy against brain tumors being limited by the ability of extracranially activated T cells to infiltrate into, and remain active in, the brain[50].

Our NGS revealed more than a million overall mutations in the B16-APOBEC3B[ACTIVE] cells compared to control cells (Supplementary Fig. 2B). Of these, ~68,000 contained the classical APOBEC signature/motif. In ongoing studies of mutagenesis of the *csde1* gene, we have observed induction of three mutations within the *csde1* gene which are detectable at levels of >20% in the

whole cell population within 48 h post transduction, during which time APOBEC3B expression was still readily detected in B16-APOBEC3B[ACTIVE] cells. All three mutations have an APOBEC mutational signature, and all three were undetectable in B16 cells transduced with the APOBEC3B[INACTIVE] vector. In contrast, we also detected one further mutation in the *csde1* gene in APOBEC3B[ACTIVE] cells. However, this fourth mutation was not detectable at any point up to 120 h post transduction—by which time we could no longer detect APOBEC3B over-expression in the cells. In contrast, this mutation was present in the population at day 21 post transduction, suggesting that it was generated subsequent to the direct mutagenic activity of APOBEC expression. None of these four mutations were detected in the

B16-APOBEC3B[INACTIVE] cells at any time points following transduction. Based on these preliminary observations, we hypothesize that the initial overexpression of APOBEC3B induces a core set of mutations, carrying the classical APOBEC3B mutational signature/motif, and often as a set of strand coordinated mutations (kategeis)[51]. With time, these core mutations, affect pathways in the cell which will, themselves, lead to further cascades of mutations throughout the genome. With time, the overall mutational activity seen in the cell population will reflect the sum of the initial core APOBEC3B mutations/kategeis, along with an amplification of the mutational burden through additional cascades of mutation resulting from the phenotypic effects of the core mutations. This hypothesis is consistent with the observed mutational activity of APOBEC in humans, where APOBEC mutagenesis has been shown to induce mutations in cancer driver genes in several different cancer types, which then drive the transition to an increasing malignant phenotype through additional mutation[9,26,40,52–54]. Moreover, APOBEC activity has been shown to drive branched evolution through the acquisition of sub-clonal mutations during the evolutionary progression of a variety of human cancers[55–57] which is consistent with the presence of the sub-clonal, APOBEC signature mutations that we detected in the B16-APOBEC3B[ACTIVE] vaccine population (Supplementary Fig. 2B).

Using next generation whole-genome sequencing to identify potential APOBEC3B-generated heteroclitic peptides, we selected ten mutant skin-expressed peptides (generated from mutations with APOBEC3B-like signatures) with increased MHC-I binding affinity. Of these ten candidates, all but one of which were identified from the NGS data to be present in the B16-APOBEC3B[ACTIVE] but not in the B16-APOBEC3B[INACTIVE] cell vaccines (Fig. 4b), one mutant peptide derived from the CSDE1 protein stimulated an ex vivo recall response against parental B16 cells. CSDE1 is a multifunctional RNA binding protein that regulates RNA translation and turnover[58–62]. B16 cell lysates expressing the APOBEC3B-altered CSDE1 peptide, but not the wild-type peptide, enhanced survival of tumor bearing mice when combined with ICB, although less effectively than the APOBEC3B-modified cell vaccine, highlighting the advantage of a wide spectrum library approach[63]. Thus, we have shown that mutated CSDE1 represents at least one of the directly APOBEC3B-mutated genes which can contribute to the immunizing potential of the B16-APOBEC3B[ACTIVE] vaccine (Fig. 5). However, our data also clearly show that the full immunizing potential of the B16-APOBEC3B[ACTIVE] vaccine is not recapitulated solely by mutated CSDE1 (Fig. 5d, e). Therefore, it is highly probable that additional mutations—both within the core APOBEC3B-mediated mutational load, as well as within the subsequent mutational amplification steps—also contribute to the protective effects seen with the vaccine. We confirmed that the mutated CSDE1 peptide was a heteroclitic epitope because splenocytes from mice treated with B16-APOBEC3B[ACTIVE]/ICB secreted IFNγ in response to both the altered, and, critically, the unaltered CSDE1 epitope, expressed in transfected TC2 prostate cells.

Our data show that APOBEC3B-induced mutations in the vaccine cell population are not clonal and can be present at frequencies of up to ~60% (Fig. 4b). This is significant because the response of a tumor to immune checkpoint blockade is dependent upon both the levels of mutation burden and the level of clonal neoantigens present. Thus, if a given neoantigen is expressed in only a proportion of the tumor, even a potent T-cell response against that neoantigen will not clear the entire tumor burden. Interestingly, the heteroclitic nature of a proportion of the neoepitopes induced by APOBEC3B mutation may help, at least partially, to overcome this problem. In this respect, if the

APOBEC3B-mutated vaccine induces several different heteroclitic neoepitopes, the T cells responding to these neoantigens will also respond to the unmutated epitopes expressed on the parental tumor cells, as we have shown here. Therefore, since it is likely that a high proportion of the tumor cells will express the unmutated epitope against which the T cell reactivity is induced, the problem of a lack of clonal expression of a mutated neoepitope within every cell of a tumor may be circumvented.

As shown in Supplementary Fig. 2, in addition to a core set of characteristic APOBEC signature mutations, multiple accessory mutations were also generated in the APOBEC[ACTIVE] cells compared to the APOBEC[INACTIVE] cells. These mutations included indels which generated frameshift mutations which might themselves generate novel antigens. It may be that such new antigens would be less significant in the vaccine strategy we describe here because the T-cell responses raised against completely novel antigens—generated by frameshift mutations—would be unlikely to cross-react back against the parental tumors in which the neoantigens are not expressed.

Finally, to support translation of APOBEC3B-modified human tumor cell vaccines into clinical trials, we demonstrated that human T cells activated in the presence of APOBEC3B[ACTIVE]-modified tumors of diverse types have increased reactivity to unmodified tumors, that this is further enhanced with ICB, and can be used in the context of pulsing human dendritic cells with APOBEC3B-modified vaccines.

Our data here show that actively driving mutation in tumor cells to enhance mutational load can be achieved using forced overexpression of APOBEC3B. This generates a genome-wide library of defined mutations, which can be tracked by their characteristic mutational signature, and associated with different phenotypes (Fig. 5 and Supplementary Fig. 3). Our data here also show that, while APOBEC3B-driven mutation decreased sensitivity to frontline chemotherapy, it simultaneously enhanced sensitivity to checkpoint blockade immunotherapy. The overall outcome was that the therapeutic gain in immunotherapy significantly outweighed the loss of responsiveness to chemotherapy. This concept opens up opportunities for a whole new approach to cancer immunotherapy in which mutagenesis is actively driven within tumors rather than inhibited. Current therapies based on identification of neoepitopes are highly individualized, time, labor, and cost intensive, and may not be readily applicable for tumors with a low mutational burden. In contrast, we describe here a readily translatable clinical method by which forced expression of APOBEC3B in the context of an ex vivo generated vaccine can induce an enhanced mutational load which combines with immune checkpoint blockade for very potent cancer immunotherapy. This is in sharp contrast to the risks of inducing tumor cell escape from therapy, which has previously been the hallmark of APOBEC3B mutagenesis. We also show that APOBEC3B-driven mutagenesis generates a genomic library of heteroclitic neoantigens that, while not expressed within the in situ tumor, can induce antitumor immunity through activation of reactive antitumor T cells, thereby significantly enlarging the scope of available vaccine targets. Studies are currently underway to determine whether APOBEC3B-mediated mutagenesis is uniquely able to induce enhanced immunogenicity of tumor cell vaccines or whether other mutagenic compounds/processes—such as chemotherapy or radiation therapy—can have similar effects.

Importantly, this HEAT will contain multiple neoepitopes of which different subsets will also be heteroclitic for each individual patient, both dispensing with the need to identify patient specific neoepitopes, and overcoming the need for pre-existing targetable mutations in the cancer genome. Nonetheless, extensive additional preclinical studies will have to be conducted to

assess the potential for autoimmune induction using such a vaccine approach. In addition, the concept of direct transfer of APOBEC3B into live tumor cells to induce immunogenicity through formation of novel heteroclitic neo-epitopes cannot yet be considered as a clinical option due to the risks of inducing further tumor promoting mutations. Finally, we show that APOBEC3B-induced heteroclitic neoepitope vaccines are effective against different tumor types, growing in different anatomic locations, indicating that this represents a technically simple, widely applicable method to exploit neoepitopes for cancer immunotherapy.

## Methods

**Study design.** This study was designed to evaluate the use of an APOBEC3B-modified vaccine to prolong survival of tumor bearing mice and investigate its synergy with immune checkpoint blockade. Seven mice per group were used for each experiment to achieve statistical power to make multiple comparisons. Mice were randomized at time of tumor implantation and tumors were measured by a single blinded individual.

**Cell lines.** B16.F1 murine melanoma cells were obtained from the ATCC. B16TK cells were derived from a B16.F1 clone transfected with a plasmid expressing the herpes simplex virus thymidine kinase (HSV-1 TK) gene in 1997/1998. Following stable selection in 1.25 μg/mL puromycin, these cells were shown to be sensitive to Ganciclovir (Cymevene) at 5 μg/ml [20–22]. B16TK cells were grown in DMEM (HyClone, Logan, UT, USA) + 10% fetal bovine serum (FBS) (Life Technologies) + 1.25 μg/mL puromycin (Sigma) until challenge. GL261 cells were obtained from Dr. Aaron Johnson (Mayo Clinic) in 2014. TRAMP-C2 (TC2) cells are derived from a prostate tumor that arose in a TRAMP mouse and were characterized as described[64,65]. Mel888 cells were obtained from the Imperial Cancer Research Fund (ICRF) in 1997/1998 and were grown in DMEM (Hyclone, Logan, UT, USA) + 10% FBS (Life Technologies). SJPDGF1 pediatric diffuse intrinsic pontine glioma (DIPG) cells were a generous gift from Dr. Cynthia Wetmore (previously at St. Jude's, now at Phoenix Children's Hospital) and were cultured in TSM media, which consists of 50% Neurobasal-A Medium, 50% DMEM/F-12, 10 mM HEPES solution, 1 mM MEM Sodium Pyruvate solution, 1× GlutaMAX Supplement, 1× Antibiotic/Antimycotic solution, 1× B-27 Supplement Minus Vitamin A, 20 ng/mL human epidermal growth factor (Shenandoah Biotech), 20 ng/mL human fibroblast growth factor basic-154 (Shenandoah Biotech), 10 ng/mL human PDGF-AA (Shenandoah Biotech), 10 ng/mL human PDGF-BB (Shenandoah Biotech), and 2 μg/mL heparin solution (StemCell Technologies). Cell lines were authenticated by morphology, growth characteristics, polymerase chain reaction (PCR) for melanoma specific gene expression (gp100, TYRP-1, and TYRP-2) and biologic behavior, tested mycoplasma-free, and frozen. Cells were cultured less than 3 months after thawing. Cells were tested for mycoplasma using the MycoAlert Mycoplasma Detection Kit (Lonza Rockland, Inc., ME, USA).

**Immune cell activation.** Spleens and lymph nodes were immediately excised from euthanized C57Bl/6 or OT-I mice and dissociated in vitro to achieve single-cell suspensions. Red blood cells were lysed with ACK lysis buffer (Sigma-Aldrich) for 2 min. Cells were resuspended at $1 \times 10^6$ cells/mL in Iscove's Modified Dulbecco's Medium (IMDM; Gibco) supplemented with 5% FBS, 1% penicillin–streptomycin, 40 μmol/L 2-Mercaptoethanol. Cells were cocultured with target cells as described in the text. Cell-free supernatants were then collected 72 h later and tested for IFNγ (Mouse IFNγ ELISA Kit; OptEIA, BD Biosciences) production by ELISA as directed in the manufacturer's instructions.

**APOBEC3 overexpression and vaccine preparation.** B16TK cells were transduced with a retroviral vector encoding either full length functional APOBEC3B (APOBEC3B$^{ACTIVE}$) or a mutated, catalytically inactive form of APOBEC3B (APOBEC3B$^{INACTIVE}$) as a negative control obtained from Dr. Reuben Harris (University of Minnesota, MN). Forty-eight hour post transduction with either pBABE-Hygro APOBEC3B$^{ACTIVE}$ or pBABE-Hygro APOBEC3B$^{INACTIVE}$ viruses, bulk populations of cells were selected in hygromycin for 2 weeks and used for experiments. Overexpression of APOBEC3B was confirmed by both Western Blot (using a rabbit monoclonal anti-human APOBEC3B (184990, Abcam, San Francisco, CA)) and qrtPCR as previously described[16]. We have observed that over-expression of APOBEC3B is toxic in that elevated levels of APOBEC3B are seen within 72 h post transfection/transduction and return to similar levels to that seen in parental unmodified cells. We believe this is because mutagenesis by APOBEC3B is tolerable to the cell up to a certain threshold, and then in cells where critical mutations are induced, this can be lethal. In other cells, overexpression of APOBEC3B may not reach the threshold, or mutations may not be induced in critical genes, allowing those cells to survive carrying the APOBEC3B-induced mutations. Consistent with this hypothesis, short term expression of APOBEC3B$^{ACTIVE}$, but

not the catalytically inactive mutant APOBEC3B$^{INACTIVE}$, led to significant levels of cell killing within between 72 and 96 h post transduction (Supplementary Fig. 4).

B16 or GL261 cells either left untreated, or modified to over-express either functional APOBEC3B (B16-APOBEC$^{ACTIVE}$) or the nonfunctional APOBEC3B INACTIVE protein (B16-APOBEC3B$^{INACTIVE}$) were expanded in T175 flasks. At 80–90% confluency, cells were trypsinized and washed three times in phosphate-buffered saline (PBS) (HyClone). Aliquots of $5 \times 10^7$ cells were resuspended in a volume of 1 ml PBS and then freeze–thawed for three cycles in liquid nitrogen. Totally, 100 μl of these freeze–thawed vaccine preparations were administered intraperitoneally (i.p.) to mice (the equivalent of $5 \times 10^6$ cells per injection).

Human cell lines were transduced with APOBEC3B or GFP using a lentiviral vector. After 2 h, the media was replaced and cells were cultured for five more days before use as APOBEC3B-modified or control vaccines.

**In vivo experiments.** All in vivo studies were approved by the Institutional Animal Care and Use Committee at Mayo Clinic. A 6–8-week-old female C57BL/6 mice were purchased from Jackson Laboratories (Bar Harbor, Maine). Mice were challenged subcutaneously with $2 \times 10^5$ B16TK murine melanoma cells, their APOBEC3B$^{ACTIVE}$ or APOBEC3B$^{INACTIVE}$ derivatives, parental B16 murine melanoma cells, or TC2 murine prostate carcinoma cells in 100 μL PBS. Alternatively, $5 \times 10^4$ GL261 cells or $1 \times 10^4$ B16 cells were implanted in 2 μL intracranially into the brainstem. Subcutaneous tumors were treated with a two week course of GCV (50 mg/kg) administered i.p. daily; or with a 5-day course of B16-APOBEC3B$^{ACTIVE}$ or B16-APOBEC3B$^{INACTIVE}$ cell vaccines as described in the text. Subcutaneous tumors were measured three times per week, and mice were euthanized when tumors reached 1.0 cm in diameter. GL261 tumor cells were stereotactically implanted into the brainstem of C57Bl/6 mice using previously published coordinates of −0.8 mm on the y plane, −1 mm on the x plane, and −5 mm on the z plane[66]. Intracranial tumors were treated with a 5-day course of GL261- or B16-APOBEC3B$^{ACTIVE}$ or APOBEC3B$^{INACTIVE}$ cell vaccines as described in the text. Mice were sacrificed upon emergence of neurological symptoms or weight loss.

**Immune cell depletions and checkpoint inhibition.** To deplete specific immune subsets, mice were treated with intraperitoneal (i.p.) injections (0.1 mg per mouse) of anti-CD8 (Lyt 2.43, BioXCell), anti-CD4 (GK1.5, BioXCell), anti-NK (anti-asialo-GM-1, Cedarlane), and IgG control (ChromPure Rat IgG, Jackson ImmunoResearch) at day 4 after tumor implantation and then weekly thereafter. Fluorescence-activated cell sorting analysis of spleens and lymph nodes confirmed subset specific depletions. For immune checkpoint blockade, mice were treated intravenously with anti-PD1 (0.25 mg; catalog no. BE0146; Bio X Cell), anti-CTLA-4 (0.1 mg; catalog no. BE0164; Bio X Cell), anti-asialo GM1 (0.1 mg; catalog no. CL8955; Cedarlane), or isotype control rat IgG (catalog no. 012-000-003; Jackson ImmunoResearch) antibody at times described in each experiment.

**Next generation sequencing.** DNA for whole-genome sequencing was prepared from B16tk cells that overexpressed either APOBEC3B$^{ACTIVE}$ or parental, unmodified cells. These cells were generated initially from an experiment investigating the role of APOBEC3B overexpression on the generation of resistance to VSV oncolysis[16]. Following three rounds of infection with VSV, genomic DNA was isolated from APOBEC3B expressing cells resistant to infection or parental, mock-infected cells. Five hundred nanograms of each sample was prepared using an Ultra kit (New England Bio) and underwent paired end 150 bp sequencing on the Illumina HiSeq 4000 by Mayo's Genome Analysis Core. Sequences were aligned to the mm10 C57Bl/6 genome by the Mayo's Bioinformatics Core Facility and mutational changes were detected between the two samples. These mutations contained 244C to T and G to A missense mutations which were unique to the APOBEC3B$^{ACTIVE}$ line. These coding variants were filtered through NET MHC 2.0 binding affinity algorithm to identify octamer or nonamer peptides whose binding affinity for H2K$^b$ or H2K$^d$ was below a threshold of 500 nM, and whose corresponding wild-type peptides had a binding affinity to the same molecules above 500 nM. A list of 36 candidates was further refined using the EMBL-EBI Expression Atlas for expression in skin tissue to identify 10 high affinity APOBEC3B- induced heteroclitic peptides.

**APOBEC3B-modified epitope expression.** cDNAs encoding either wild type (unmutated) or APOBEC3B-mutated 9 mer peptides derived from the screen described above were synthesized and cloned into the plasmid pcDNA3.1 + P2A-eGFP (Genescript), where the cDNA is expressed from a CMV promoter and co-expressed with an eGFP protein. pcDNA3.1-EPITOPE + P2A-eGFP plasmids were transfected into B16 cells using Lipofectamine. Cultures in which transfection of over 50% of cells was confirmed by GFP analysis 48–72 h post transfection were used either as targets in immune activation assays (above) or were expanded in T175 flasks. At 80–90% confluency, cells were trypsinized and washed three times in PBS. Aliquots of $5 \times 10^7$ cells were resuspended in a volume of 1 ml PBS and then freeze–thawed for three cycles in liquid nitrogen. Totally, 100 μl of these freeze–thawed vaccine preparations were administered intraperitoneally to mice (the equivalent of $5 \times 10^6$ cells per injection).

**Human T cell in vitro education and restimulation**. Fresh PBMCs from a healthy donor were acquired from the Mayo Clinic Blood Bank. CD3+ T cells were isolated using a magnetic sorting kit (Miltenyi Biotech) and activated using CD3/CD28 beads (ThermoFisher). T cells were immediately cocultured at a ratio of 10:1 with interferon gamma pretreated (200 U/mL for 12 h) APOBEC3B or GFP-transduced Mel888 cells. After 10 days of coculture, CD3+ T cells were re-isolated using a magnetic sorting kit (Miltenyi Biotech) and labeled with Cell Trace Violet (ThermoFisher) for proliferation analysis. These T cells were then cocultured with interferon gamma pretreated (200 U/mL for 12 h) parental Mel888 cells for 72 h, followed by interferon gamma ELISA (R&D), flow cytometric proliferation analysis via cell trace violet dilution, and counting of remaining Mel888 cells.

For vaccination with SJPDGF1 tumors, T cells were isolated and activated as described above. Autologous monocyte-derived dendritic cells were matured by isolating CD14+ cells by magnetic sorting (Miltenyi Biotech), followed by incubation with human granulocyte-macrophage colony-stimulating factor (GM-CSF) (800 U/mL) and IL-4 (1000 U/mL). On Days 3 and 5, media was replaced with human GM-CSF (1600 U/mL) and IL-4 (1000 U/mL). On Day 7, non-adherent cells were collected, washed with PBS, and resuspended in medium containing GM-CSF (800 U/mL), IL-4 (1000 U/mL), TNF-alpha (1100 U/mL), IL-1beta (1870 U/mL), IL-6 (1000 U/mL), and PGE2 (1 μg/mL). Two days later, dendritic cells were harvested for co-incubation with activated T cells at a ratio of 1:10. On each of the first 3 days of coculture, SJPDGF1-GFP, Mel888-GFP, or SJPDGF1-APOBEC3B cell lysates were added to the coculture. Ten days after initial coculture, CD3+ T cells were isolated using magnetic bead sorting (Miltenyi Biotech), cocultured with newly matured monocyte-derived dendritic cells, and fed with parental SJPDGF1 lysate. Three days later, supernatant was collected for interferon gamma ELISA (R&D).

**Cytidine deamination assay**. Lysates of un-transduced (UTD) parental, APOBEC3B$^{ACTIVE}$ or APOBEC3B$^{INACTIVE}$ cells were incubated for 2 h at 37 °C with a 43 bp fluorescein labeled probe. APOBEC3B present in the lysate will deaminate the single C to a U and in the presence of 6N NaOH the U-containing substrate will be cleaved into 30 bp (labeled) and 12 bp (unlabeled) products which could be visualized on a 15% TBE-Urea polyacrylamide gel electrophoresis gel[67].

**Statistics**. Survival curves were analyzed by the Log-Rank test with Holm–Bonferroni correction for multiple comparisons. Student's $t$ tests, one-way ANOVA and two-way ANOVA were applied for in vitro assays as appropriate. Statistical significance was set at $p < 0.05$ for all experiments.

**Primer sequences**. CSDE1* sequencing 5′-TCACGAAGTGCTGCTGAAGT-3′ APOBEC3B qRT-PCR Forward 5′-ATGAATCCACAGATCAGAAATCCG-3′ APOBEC3B qRT-PCR Reverse 5′-GGTAATCTCTTTCCCAGTAGTAGT-3′

**Reporting summary**. Further information on research design is available in the Nature Research Reporting Summary linked to this article.

## Data availability

All relevant data are available in the Article, Supplementary Information or from the corresponding author upon request. Binary sequence alignments/map files can be accessed using the NIH Sequence Read Archive: SRP159367.

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

## Acknowledgments

The authors thank Toni L. Woltman for expert secretarial assistance. They would additionally like to thank the Genome Analysis and Bioinformatics Core Facilities for their help in obtaining and interpreting the sequencing results, and John Smestad for assistance in uploading raw sequencing data. This work was funded in part by The European Research Council, The Richard M. Schulze Family Foundation, the Mayo Foundation, Cancer Research UK, the National Institute of Health (R01CA175386 and R01CA108961), The University of Minnesota and Mayo Clinic Partnership, a grant from Terry and Judith Paul, The Shannon O'Hara Foundation, Hyundai Hope On Wheels, and a research grant from Oncolytics Biotech Inc.

## Author contributions

R.G.V., T.K., M.S., C.B.D., L.E., R.S.H., J.S.P., A. Mo, K.G.S., A. Me., K.H., P.S. and H.P. conceived and designed all studies. T.K., J.M.Th. and J.M.To. developed all of the methodology. R.G.V., T.K., J.M.Th., J.M.To. and A. Mi. acquired all of the data. R.G.V., T.K., C.B.D, M.S, L.E., A.L.H., K.G.S. and P.W. analyzed and interpreted all of the data. R.G.V., C.B.D., M.S., L.E., A.L.H., J.S.P., P.W., T.K., J.M.To., P.S., A.S., H.P., K.H. and A. Me. wrote, reviewed, and/or revised the paper. R.S.H. and C.W. provided administrative, technical, or material support for the study and R.G.V. supervised the study.

## Competing interests

R.G.V., L.E., T.K. and A.L.H. have submitted a patent pertaining to this work.
