## [Peer Review File · Nature Communications]

Reviewers' comments:

Reviewer #1 (Remarks to the Author):

The authors propose an interesting concept to improve the immunogenicity of cancer cells by introducing APOBEC into the cells to induce mutations. They demonstrate that this method will provide a potent vaccine in murine tumors and show as a proof of concept that human T cells can also be stimulated effectively in vitro. The mutated epitopes are heteroclitic in that they induce a response to unmutated epitopes on the parental tumor as well.

Comments:

1. The significant impact of the approach of the paper is as a vaccine method. However, the authors imply in the introduction and discussion (and because the first experiment involves using the method in a live tumor in a mouse) that they are perhaps advocating a highly controversial idea that APOBEC treatment of tumors might be possible (akin to recommending smoking for patients with lung cancer or more sun for patients with melanoma) in order to increase mutations and therefore a better response to immunotherapy. I think this sort of implication should be avoided.

2. Do the mutations affect MHC or other antigen presentation machinery and if so, what is the implication?

3. Among the numerous mutations were indels found that yielded frameshifts that could generate important new antigens?

4. The heteroclitic response is shown for the WT tumors and is important. But what if among the thousands of neoantigens are heteroclitic peptides for normal self presented peptides on normal cells? In a typical vaccine defined peptides are often used. In this case, the repertoire may be large and ill-defined. This should be discussed as a risk.

5. Only the size of tumor was indicated to sacrifice mice. What if pseudoprogession led to increase over 1 cm? Other papers where they use B16 melanoma and ICB use 1.5 or even 2 cm³ as threshold. Perhaps Figure 1 could have show tumor growth as well and not just survival to rule out pseudoprogession.

6. In fig 1B the difference between active and inactive survival is based on death of 2 mice. This may not be significant. In Fig 2 the stats should be compared between the active and inactive groups (not vs the no vaccine group.)

7. The discussion is long, and especially the last paragraph redundant.

Reviewer #2 (Remarks to the Author):

Driscoll et al. demonstrate that elevated mutation burden driven by the overexpression of APOBEC3B in both melanoma, glioblastoma, and prostate tumour cell line transplant models leads to more rapid resistance development to chemotherapy, but also increased immune checkpoint

blockade sensitivity and an increased anti-tumor vaccination response. The manuscript is well written and addresses a number of questions raised in recent literature in regards to levels of mutation burden and response to immune checkpoint blockade. The manuscript is novel, but I feel the authors could characterise the nature of the increased mutation burden further and therefore recommend major revisions. Below are comments for the author to consider:

Major comments:

1. The authors identified over 1,000,000 mutations or indels unique to B16-APOBEC3BActive overexpressing cells, with 244 of those containing missense mutations consistent with APOBEC3B mutational activity. Could the authors please characterise the clonality of the APOBEC mutations in the cell population, as it has been shown in the literature that immune checkpoint blockade responsiveness is affected by not only the levels of mutation burden but also the level of clonal neoantigens present.

2. It is strange that of 1,000,000+ mutations, that are supposedly generated by APOBEC3B, only 244 contain the classical APOBEC mutational signature/motif. Could the authors characterise the other mutations that were generated (mutational signature, clonality, binding affinity etc)? How can this be explained and how does this compare to APOBEC mutagenesis in humans? The authors need to prove that their model is representative for APOBEC mutagenesis. Could the remaining 1,000,000+ mutations perhaps convey the protective effect as opposed to the 244 mutations?

3. The claim of antigen spreading seems weak, the CSDE1 mutation the authors focus on a lot, seems to be only detected by Sanger sequencing. The authors should use NGS or digital droplet PCR to validate its presence in the APOBEC-treated vaccine and absence in the untreated vaccine. Particularly as the authors claim its subclonal in the APOBEC-treated vaccine, Sanger lacks sensitivity to determine this.

4. Could the authors please demonstrate that B16-APOBEC3BInactive overexpressing cells don't also have increased mutations or indels in comparison to the B16-APOBEC3BActive overexpressing cells.

5. The authors state that overexpressing APOBEC3B is toxic to some cells as mutagenesis is tolerated only up to a certain threshold or induces mutations in critical genes. Could the authors please demonstrate using a cell death assay that this is the case? It is important to assess the levels of cell death to understand if the transfected/transduced population is bottlenecked to only a few clonal cells.

6. Is the protective effect of the APOBEC3B-modified tumor vaccine due to APOBEC3B-mediated mutations or would any mutagenic compound/process have a protective effect?

Minor comments:

1. In the abstract and introduction please list the specific tumour types that simultaneously showed increased resistance to chemotherapy and simultaneously heightened sensitivity to immune checkpoint blockade.
2. Please define the HSVtk suicide gene on page 7 of the results section.
3. In Figure 2b and c, B16-APOBEC3BActive/anti-PD1 Treating B16 and B16-APOBEC3BActive/anti-PD1 Treating TC2 are difficult to tell apart. Please change the color or shape to make the figure clearer.
4. Please make sure to add how many times each experiment was repeated to the figures.
5. Is there a significant change in Fig1B between APOBEC3BACTIVE/GCV/aCTLA4 and APOBEC3BINACTIVE/GCV/aCTLA4? Most of the stats is omitted in Figure1A. Please add this to the figure and text.
6. Please add the P-values in the text whenever comparisons are made.
7. The authors need to be more careful inflating the clinical potential from their data - I imagine a whole tumour cell vaccine approach, with upregulation of abopec, would be highly challenging/controversial. They should acknowledge a lot more pre-clinical work would be needed on this, rather than immediate translation to humans.

Reviewer #3 (Remarks to the Author):

The authors in this manuscript demonstrate APOBEC3B mediated immune activation in B16 melanoma and GL261 glioma mice models using APOBEC3B activated vaccination strategy in combination with immune checkpoint inhibitors of T cells. The data in these two mouse models clearly suggest APOBEC3B associated tumor regression in combination with immune checkpoint inhibitors specifically associated with the APOBEC3B vaccine. Authors further demonstrate that the APOBEC3 vaccine effect in relation to neoepitopes resulted from APOBEC3B mutational signature, especially with CSDE1 minimal epitope.

However, authors' claim of CSDE1 epitope as a heteroclitic neoepitope requires further explanation/exploration. The IFN-gamma secretion by splenocyte from APOBEC3B-ACTIVE treated mice in the presence of PD1-checkpoint inhibitor when stimulated with wild type epitopes, expressed by B16 cells, is similar to B16 cells expressing no epitopes (Figure 4D). Thus, the activation of T cells could be associated with additional neoepitopes reactive T cells and not with wild type minimal epitopes reactive to T cell activated by CSDE1 neoepitope as claimed by the authors. This is also evident from the data in Figure 5D and E, that CSDE1* (neoepitope) seem to play a minor role in the tumor rejection capacity compared to the complete APOBEC3B-ACTIVE vaccine. If CSDE1 was the driver for tumor recognition (serving as a heteroclitic epitope) then it must have generated an effect similar to the complete vaccine (APOBEC3B-ACTIVE).

Thus, the role of CSDE1 such be clarified and the conclusion be modified.

The authors further validate the APOBEC3B mediated vaccine approach on human tumor samples, which can potentially be translated to clinical application. Altogether, this manuscript provide detailed and effective approach for cancer immunotherapy using APOBEC3B modification.

RESPONSE TO REVIEWERS

Reviewer #1:

The authors propose an interesting concept to improve the immunogenicity of cancer cells by introducing APOBEC into the cells to induce mutations. They demonstrate that this method will provide a potent vaccine in murine tumors and show as a proof of concept that human T cells can also be stimulated effectively in vitro. The mutated epitopes are heteroclitic in that they induce a response to unmutated epitopes on the parental tumor as well.

1. The significant impact of the approach of the paper is as a vaccine method. However, the authors imply in the introduction and discussion (and because the first experiment involves using the method in a live tumor in a mouse) that they are perhaps advocating a highly controversial idea that APOBEC treatment of tumors might be possible (akin to recommending smoking for patients with lung cancer or more sun for patients with melanoma) in order to increase mutations and therefore a better response to immunotherapy. I think this sort of implication should be avoided.

The Reviewer's point is well taken. We agree with the Reviewer that translation of an APOBEC3B-modified cell vaccine, along with immune checkpoint blockade, is readily translatable because the vaccine cells will be non-viable - as was the case in our therapy experiments here. However, the proposal to transfer APOBEC directly into tumors to induce enhanced immunogenicity through the generation of heteroclitic neo-epitopes would be highly controversial and would require co-expression of a cytotoxic gene to ensure eradication of any APOBEC3B-mutated cells which could, as a result, become more malignant. Therefore, as requested by the Reviewer, we have added the following text to the **Discussion** on **page 21**:

Importantly, this Heteroclitic Epitope Activated Therapy (HEAT) will contain multiple neoepitopes of which different subsets will also be heteroclitic for each individual patient, both dispensing with the need to identify patient specific neoepitopes, and overcoming the need for pre-existing targetable mutations in the cancer genome. **Nonetheless, extensive additional pre-clinical studies will have to be conducted to assess the potential for autoimmune induction using such a vaccine approach. In addition, the concept of direct transfer of APOBEC3B into live tumor cells to induce immunogenicity through formation of novel heteroclitic neo-epitopes cannot yet be considered as a clinical option due to the risks of inducing further tumor promoting mutations.**

2. Do the mutations affect MHC or other antigen presentation machinery and if so, what is the implication?

Our NGS data indicated 1066 mutational changes between the APOBEC3B^{ACTIVE} and APOBEC3B^{INACTIVE} cells, resulting in a total of 7 amino acid changes in four genes of

antigen presentation pathways. Five of these mutations occurred at low clonality within the sequenced populations (a frequency of <8%). In contrast, two mutations which were detected in the non-classical major histocompatibility class I genes H2-M1 and H2-M2 occurred at a relatively high frequency within the sequenced population (36 and 52% respectively). APOBEC3B-induced mutations in MHC molecules, or other molecules of the antigen presentation pathway, may significantly affect the ability of the modified mutated cells to present their antigens. We do not believe that this would be a major impediment to efficacy in the vaccine setting, as we hypothesize that the majority of the (heteroclitic) neo-epitope antigen presentation will be through professional APC which cross present the relevant neo-epitopes to potentially reactive T cells. However, APOBEC3B mutation of the MHC molecules/antigen presenting pathways in tumor cells which are still potentially able to present available epitopes/neo-epitopes could reduce the efficacy of any T cell mediated immunotherapy. Our experiments in **Figure 1** suggest that if this occurred in the APOBEC3B^{ACTIVE} tumors it was not widespread enough to prevent recognition of the tumors by anti-tumor T cells primed against the mutated tumors. Nonetheless, the possible mutation of MHC/antigen presentation by APOBEC3B would be another strong argument about the risks of direct transfer of APOBEC3B into live tumor cells to induce immunogenicity (see point 1 above).

3. Among the numerous mutations were indels found that yielded frameshifts that could generate important new antigens?

Our NGS data show that multiple indels were present in the APOBEC^{ACTIVE} vaccine cells and we have added this new data as a new **Table S2B**. We have added the following text on **page 10** of the **Results**:

A whole genome sequencing screen of the B16tk-APOBEC3B^{ACTIVE}-modified, VSV-escaped population, compared to B16tk parental cells, identified over 1,000,000 mutations or indels unique to the B16-APOBEC3B^{ACTIVE} overexpressing cells (**SRA Submission: SRP159367**). Using an *in vitro* cytidine deamination assay (**Figure S2A**), we confirmed that, whereas the APOBEC3B^{ACTIVE} vector induced cytidine deaminase activity consistent with APOBEC mutagenesis, the APOBEC3B^{INACTIVE} vector did not. Consistent with these data, whilst we detected APOBEC3B-signature mutations in the B16 APOBEC3B^{ACTIVE} cells, these were not, or very rarely, present in the corresponding B16 APOBEC3B^{INACTIVE} cells (**Fig.4B**). However, in addition to core, APOBEC3B signature mutations, multiple accessory mutations were also generated in the APOBEC^{ACTIVE} cells compared to the APOBEC^{INACTIVE} cells, including nucleotide changes and indels which occurred typically at low clonalities in the vaccine cells (**Figure S2B**). Of these total mutations detected by NGS, 244 contained C to T or G to A transitions that lead to missense mutations consistent with APOBEC3B mutational activity (**Table S1**)³⁷.

In addition, we have added the following text to the **Discussion** on **page 20**:

As shown in **Figure S2**, in addition to a core set of characteristic APOBEC signature mutations, multiple accessory mutations were also generated in the APOBEC^{ACTIVE} cells compared to the APOBEC^{INACTIVE} cells. These mutations included indels which generated frameshift mutations

which might themselves generate novel antigens. It may be that such new antigens would be less significant in the vaccine strategy we describe here because the T cell responses raised against completely novel antigens – generated by frameshift mutations – would be unlikely to cross react back against the parental tumors in which the neo-antigens are not expressed.

4. The heteroclitic response is shown for the WT tumors and is important. But what if among the thousands of neoantigens are heteroclitic peptides for normal self presented peptides on normal cells? In a typical vaccine defined peptides are often used. In this case, the repertoire may be large and ill-defined. This should be discussed as a risk.

The reviewer's point is well made. To address this comment, we have added the following to the **Discussion** on **page 17**:

Consistent with this hypothesis, the modest survival enhancement generated by the B16-APOBEC3B^{ACTIVE} vaccine in the absence of additional adjuvants was converted into a curative treatment when combined with ICB, dependent upon CD4, CD8 and NK cells. Interestingly, even in the presence of ICB to overcome immunosuppressive mechanisms acting on the activated anti-tumor T cell response⁴⁵⁻⁴⁸, we did not observe any signs of autoimmunity. **However, in each type of tumor which might be treated by this approach, detailed pre-clinical histological and toxicological studies should be performed to confirm that heteroclitic T cell responses against APOBEC3B-mutated neo-epitopes do not raise autoreactive T cell activity against critically important self-peptide epitopes expressed on the corresponding normal tissues.**

5. Only the size of tumor was indicated to sacrifice mice. What if pseudoprogession led to increase over 1 cm? Other papers where they use B16 melanoma and ICB use 1.5 or even 2 cm³ as threshold. Perhaps Figure 1 could have show tumor growth as well and not just survival to rule out pseudoprogession.

As requested by the Reviewer, we have now added a new **Figure 1C** showing tumor growth curves.

6. In fig 1B the difference between active and inactive survival is based on death of 2 mice. This may not be significant. In Fig 2 the stats should be compared between the active and inactive groups (not vs the no vaccine group.)

We have added the statistics as requested to **Figure 1B** and **Figures 2C&D**.

7. The discussion is long, and especially the last paragraph redundant.

We have reduced the length of the Discussion in the original manuscript by about 450 words and have removed the final paragraph as requested by the Reviewer.

Reviewer #2:

Driscoll et al. demonstrate that elevated mutation burden driven by the overexpression

of APOBEC3B in both melanoma, glioblastoma, and prostate tumour cell line transplant models leads to more rapid resistance development to chemotherapy, but also increased immune checkpoint blockade sensitivity and an increased anti-tumor vaccination response. The manuscript is well written and addresses a number of questions raised in recent literature in regards to levels of mutation burden and response to immune checkpoint blockade. The manuscript is novel, but I feel the authors could characterise the nature of the increased mutation burden further and therefore recommend major revisions. Below are comments for the author to consider:

We thank the Reviewer for these positive comments and have addressed the points raised below.

1. The authors identified over 1,000,000 mutations or indels unique to B16-APOBEC3B^{Active} overexpressing cells, with 244 of those containing missense mutations consistent with APOBEC3B mutational activity. Could the authors please characterise the clonality of the APOBEC mutations in the cell population, as it has been shown in the literature that immune checkpoint blockade responsiveness is affected by not only the levels of mutation burden but also the level of clonal neoantigens present.

Of the mutations which we identified with the APOBEC signature none were clonal in the vaccine cell population. Of the mutations that we selected as generating better MHC binding neo-epitopes compared to the wild type epitopes (**Figure 4**) the clonal frequencies varied between ~60% (C77080) to <10% (Ahctf1). Of those selected mutations, only one (Stag2) appeared in the APOBEC^{INACTIVE} cell population; all the others were associated with *de novo* expression of APOBEC^{ACTIVE}. We have added these new data as a new **Fig.4B**, with the following text added to the **Results** on **page 11**:

By refining this list using the EMBL-EBI Expression Atlas, eight candidate proteins were identified that were expressed in the skin: Ahctf1, C77080, Csde1, Fcgbp, Plbd2, Smc4, Stag2, and Xpo1 (which contained three candidate peptide sequence variants) (**Fig. 4A**). **Of these APOBEC-characteristic mutations, none were present clonally in the B16-APOBEC^{ACTIVE} vaccine population (Fig.4B) and varied in frequency between ~60% to <10%. Only one of the mutations (Stag2) could be detected by NGS to be present in the ABPOC3B^{INACTIVE} cells, whereas all of the others were induced by *de novo* APOBEC3B activity.**

In addition, we have added the following text to the **Discussion** on **page 19**:

We confirmed that the mutated CSDE1 peptide was a heteroclitic epitope because splenocytes from mice treated with B16-APOBEC3B^{ACTIVE}/ICB secreted IFN γ in response to both the altered, and, critically, the unaltered CSDE1 epitope, expressed in transfected TC2 prostate cells. **Our data show that APOBEC3B induced mutations in the vaccine cell population are not clonal and can be present at frequencies of up to ~60% (Fig.4B). This is significant because the response of a tumor to immune checkpoint blockade is dependent upon both the levels of mutation burden and the level of clonal neoantigens present. Thus, if a given neoantigen is**

expressed in only a proportion of the tumor, even a potent T cell response against that neoantigen will not clear the entire tumor burden. Interestingly, the heteroclitic nature of a proportion of the neoepitopes induced by APOBEC3B mutation may help, at least partially, to overcome this problem. In this respect, if the APOBEC3B-mutated vaccine induces several different heteroclitic neoepitopes, the T cells responding to these neoantigens will also respond to the unmutated epitopes expressed on the parental tumor cells, as we have shown here. Therefore, since it is likely that a high proportion of the tumor cells will express the unmutated epitope against which the T cell reactivity is induced, the problem of a lack of clonal expression of a mutated neoepitope within every cell of a tumor may be circumvented.

2. It is strange that of 1,000,000+ mutations, that are supposedly generated by APOBEC3B, only 244 contain the classical APOBEC mutational signature/motif. Could the authors characterise the other mutations that were generated (mutational signature, clonality, binding affinity etc)? How can this be explained and how does this compare to APOBEC mutagenesis in humans? The authors need to prove that their model is representative for APOBEC mutagenesis.

The Reviewer makes an excellent point and we apologize that we did not explain our thinking clearly enough in the original manuscript. We hypothesize that the initial over-expression of APOBEC3B induces a core set of mutations, carrying the classical APOBEC3B mutational signature/motif, and often as a set of strand coordinated mutations (kategeis). However, with time, these core mutations, will inevitably affect genes and processes in the cell which will, themselves, lead to further cascades of mutations throughout the genome. With time, the overall mutational activity seen in a cell population will reflect the sum of the initial core APOBEC3B mutations/kategeis, along with additional waves of mutation which result from the phenotypic effects of the core mutations. Therefore, we would propose that a relatively few core APOBEC3B-induced mutations will actually promote a greatly expanded and amplified profile of mutations both within any given cell, and across populations of cells. This hypothesis is consistent with the observed mutational activity of APOBEC in humans, where APOBEC mutagenesis has been shown to induce mutations in cancer driver genes in several different cancer types, which would then drive the transition to an increasing malignant phenotype through additional mutation. In addition, APOBEC activity in human tumors has been shown to drive branched evolution through the acquisition of sub-clonal mutations during the evolutionary progression of a variety of human cancers, which is consistent with the presence of the sub-clonal, APOBEC signature mutations that we detected in the B16 APOBEC3B^{ACTIVE} vaccine population.

We are currently testing this Core and Cascade model of mutational activity of APOBEC3B in our vaccine cells by sampling the transduced cells with time for both number, and nature, of mutations (Core, APOBEC3B characteristic; or Cascade, secondary).

Therefore, as requested by the Reviewer, we have now added a new **Fig.S2B** characterizing the other mutations that we observed through NGS of the B16 APOBEC3B^{ACTIVE} vaccine cells, along with the following text on **page 10** of the **Results**:

A whole genome sequencing screen of the B16tk-APOBEC3B^{ACTIVE}-modified, VSV-escaped population, compared to B16tk parental cells, identified over 1,000,000 mutations or indels unique to the B16-APOBEC3B^{ACTIVE} overexpressing cells (SRA Submission: SRP159367). Using an *in vitro* cytidine deamination assay (Figure S2A), we confirmed that, whereas the APOBEC3B^{ACTIVE} vector induced cytidine deaminase activity consistent with APOBEC mutagenesis, the APOBEC3B^{INACTIVE} vector did not. Consistent with these data, whilst we detected APOBEC3B-signature mutations in the B16 APOBEC3B^{ACTIVE} cells, these were not, or very rarely, present in the corresponding B16 APOBEC3B^{INACTIVE} cells (Fig.4B). However, in addition to core, APOBEC3B signature mutations, multiple accessory mutations were also generated in the APOBEC^{ACTIVE} cells compared to the APOBEC^{INACTIVE} cells, including nucleotide changes and indels which occurred typically at low clonalities in the vaccine cells (Figure S2B). Of these total mutations detected by NGS, 244 contained C to T or G to A transitions that lead to missense mutations consistent with APOBEC3B mutational activity (Table S1)³⁷.

In addition, to clarify our thinking on how to explain the spectrum of mutations observed as requested by the Reviewer, we have added the following text to the Discussion on page 17:

B16 flank tumor model, but not in the GL261 intracranial model, is consistent with ICB therapy against brain tumors being limited by the ability of extra-cranially activated T cells to infiltrate into, and remain active in, the brain⁵².

Our NGS revealed more than a million overall mutations in the B16 APOBEC3B^{ACTIVE} cells compared to control cells (Fig.S2B), of which a minority contained the classical APOBEC signature/motif. We hypothesize that the initial over-expression of APOBEC3B induces a core set of mutations, carrying the classical APOBEC3B mutational signature/motif, and often as a set of strand coordinated mutations (kategeis)⁵¹. With time, these core mutations, affect pathways in the cell which will, themselves, lead to further cascades of mutations throughout the genome. With time, the overall mutational activity seen in the cell population will reflect the sum of the initial core APOBEC3B mutations/kategeis, along with an amplification of the mutational burden through additional cascades of mutation resulting from the phenotypic effects of the core mutations. This hypothesis is consistent with the observed mutational activity of APOBEC in humans, where APOBEC mutagenesis has been shown to induce mutations in cancer driver genes in several different cancer types, which then drive the transition to an increasing malignant phenotype through additional mutation^{9, 26, 40, 52, 53, 54}. Moreover, APOBEC activity has been shown to drive branched evolution through the acquisition of subclonal mutations during the evolutionary progression of a variety of human cancers^{55, 56, 57} which is consistent with the presence of the sub-clonal, APOBEC signature mutations that we detected in the B16 APOBEC3B^{ACTIVE} vaccine population (Fig. S2B).

To address the Reviewer's point experimentally, we have now added a new Figure S2A confirming that transduction of vaccine cells with the APOBEC3B^{ACTIVE}, but not the APOBEC3B^{INACTIVE}, vector does indeed induce cytidine deaminase activity representative of APOBEC mutagenesis. Therefore, we have added the new text to the Results on page 10 as described above.

Could the remaining 1,000,000+ mutations perhaps convey the protective effect as opposed to the 244 mutations?

We absolutely agree with the Reviewer that it is highly likely that the full immunizing potential of the APOBEC3B-modified vaccine cells is contributed by a subset of both directly APOBEC3B-induced mutations (core mutations), as well as by a subset of additional mutations induced as a result of the cascade of mutations that we believe result from the activity of these core mutations. To address this point, we have added the following text to **page 19** of the **Discussion**:

B16 cell lysates expressing the APOBEC3B-altered CSDE1 peptide, but not the wild-type peptide, enhanced survival of tumor bearing mice when combined with ICB, although less effectively than the APOBEC3B-modified cell vaccine, highlighting the advantage of a wide spectrum library approach⁵⁸. Thus, we have shown that mutated CSDE1 represents at least one of the directly APOBEC3B-mutated genes which can contribute to the immunizing potential of the B16 APOBEC3B^{ACTIVE} vaccine (**Fig.5**). However, our data also clearly show that the full immunizing potential of the B16 APOBEC3B^{ACTIVE} vaccine is not recapitulated solely by mutated CSDE1 (**Figs.5D&E**). Therefore, it is highly probable that additional mutations – both within the core APOBEC3B-mediated mutational load, as well as within the subsequent mutational amplification steps – also contribute to the protective effects seen with the vaccine.

3. The claim of antigen spreading seems weak, the CSDE1 mutation the authors focus on a lot, seems to be only detected by Sanger sequencing. The authors should use NGS or digital droplet PCR to validate its presence in the APOBEC-treated vaccine and absence in the untreated vaccine. Particularly as the authors claim its subclonal in the APOBEC-treated vaccine, Sanger lacks sensitivity to determine this.

We apologize for our lack of clarity in the original manuscript. The CSDE1 mutation was indeed originally identified from our NGS experiments comparing the B16 APOBEC3B^{ACTIVE} and B16 APOBEC 3B^{INACTIVE} cell lines. As now added to the manuscript in response to point 1 above, those studies show that the CSDE1 mutation is not clonal and is present at a frequency of about 13%. To address the Reviewer's point here, we have added the following text to the **Discussion** on **page 18**:

Of these 10 candidates, all but one of which were identified from the NGS data to be present in the B16-APOBEC3B^{ACTIVE} but not in the B16 APOBEC3B^{INACTIVE} cell vaccines (**Fig.4B**), one mutant peptide derived from the CSDE1 protein stimulated an *ex vivo* recall response against parental B16 cells.

4. Could the authors please demonstrate that B16-APOBEC3BInactive overexpressing cells don't also have increased mutations or indels in comparison to the B16-APOBEC3BActive overexpressing cells.

To address the Reviewer's point here we have now added a new **Fig.4B** in which we show that, with the exception of the *Stag2* gene, B16 APOBEC3B^{ACTIVE} cells contain mutated sequences in the target genes identified from our MHC binding screen whereas

B16 APOBEC3B^{INACTIVE} cells do not. In addition, we have added a new **Figure S2A** in which we confirm (as also requested in Point 2 above) that infection of cells with the APOBEC3B^{ACTIVE} vector induces the predicted cytidine deaminase activity of APOBEC3B, whereas the APOBEC3B^{INACTIVE} vector is devoid of this enzymatic activity. Therefore, we have added the following text to the **Results** on **page 10**:

A whole genome sequencing screen of the B16tk-APOBEC3B^{ACTIVE}-modified, VSV-escaped population, compared to B16tk parental cells, identified over 1,000,000 mutations or indels unique to the B16-APOBEC3B^{ACTIVE} overexpressing cells (**SRA Submission: SRP159367**). Using an *in vitro* cytidine deamination assay (**Figure S2A**), we confirmed that, whereas the APOBEC3B^{ACTIVE} vector induced cytidine deaminase activity consistent with APOBEC mutagenesis, the APOBEC3B^{INACTIVE} vector did not. Consistent with these data, whilst we detected APOBEC3B-signature mutations in the B16 APOBEC3B^{ACTIVE} cells, these were not, or very rarely, present in the corresponding B16 APOBEC3B^{INACTIVE} cells (**Fig.4B**). However, in addition to core, APOBEC3B signature mutations, multiple accessory mutations were also generated in the APOBEC^{ACTIVE} cells compared to the APOBEC^{INACTIVE} cells, including nucleotide changes and indels which occurred typically at low clonalities in the vaccine cells (**Figure S2B**). Of these total mutations detected by NGS, 244 contained C to T or G to A transitions that lead to missense mutations consistent with APOBEC3B mutational activity (**Table S1**)³⁷.

5. The authors state that overexpressing APOBEC3B is toxic to some cells as mutagenesis is tolerated only up to a certain threshold or induces mutations in critical genes. Could the authors please demonstrate using a cell death assay that this is the case? It is important to assess the levels of cell death to understand if the transfected/transduced population is bottlenecked to only a few clonal cells.

As requested by the Reviewer, we have now added a new **Figure S4**. These data show that cells transduced with the APOBEC3B^{ACTIVE} vector grow normally for the first 72hrs post transduction (and possibly even slightly faster) but then undergo an apparent crisis period during which a significant number of cells die compared to either GFP-, or APOBEC3B^{INACTIVE}-transduced cells which lack APOBEC3B's cytidine deaminase activity (new **Figure S2A**). However, when the surviving cells were re-plated, they grew at a similar rate to the GFP-, or APOBEC3B^{INACTIVE}-transduced cells. Therefore, we have added the following text to the **Methods** on **page 25**:

We have observed that over-expression of APOBEC3B is toxic in that elevated levels of APOBEC3B are seen within 72 hours post-transfection/transduction and return to similar levels to that seen in parental unmodified cells. We believe this is because mutagenesis by APOBEC3B is tolerable to the cell up to a certain threshold, and then in cells where critical mutations are induced, this can be lethal. In other cells, overexpression of APOBEC3B may not reach the threshold, or mutations may not be induced in critical genes, allowing those cells to survive carrying the APOBEC3B-induced mutations. **Consistent with this hypothesis, short term expression of APOBEC3B^{ACTIVE}, but not the catalytically inactive mutant APOBEC3B^{INACTIVE}, led to significant levels of cell killing within between 72 and 96hrs post transduction (Figure S4).**

Figure S4. APOBEC3B overexpression is toxic to tumor cells. **A.** 10^4 B16tk cells were infected with retroviral vectors pBabe GFP; pBabeAPOBEC^{ACTIVE} or pBabeAPOBEC^{INACTIVE} at an MOI of ~10. 24hrs later, infected cells were harvested, washed and re-plated at 10^3 cells per well in triplicates. Cell growth was monitored over a period of 120hrs as shown. **B.** At the 120hr timepoint, surviving cells were harvested and pooled from the triplicates of each treatment. 10^3 cells were re-plated in triplicates and cell growth measured over a further period of 120hrs as shown.

6. *Is the protective effect of the APOBEC3B-modified tumor vaccine due to APOBEC3B-mediated mutations or would any mutagenic compound/process have a protective effect?*

This is an excellent point and one which we are actively investigating – but to which we do not yet know the answer. As the Reviewer well knows, other mutagenic treatments are associated with generation of immunogenicity *in vivo* – such as the abscopal effects of radiation and the effects of immunogenic cell death induced by some chemotherapies. Relatively few studies have been carried out on whether these effects are associated with mutation of the target antigens in the treated tumor cells in addition to the immunogenicity associated with *in vivo* cell killing, antigen release and T cell priming against the unaltered tumor antigens (self- or neo-antigens). We do know that an irradiated B16 cell vaccine is not protective against B16 tumours, either with or without checkpoint inhibitor therapy, in the regimen that we use in the current studies. We have also shown that chemotherapy-killed GL261 cells (killed by temozolomide) are not superior immunogens to unmodified tumor cells. It is possible, however, that the dose of radiation, or drug, may be critical in determining the pattern and burden of mutational load induced in the tumor cells and further dosing studies will help to determine that. We are currently screening these alternative mutagenic treatments, relative to APOBEC3B overexpression, for their ability to induce immunizing mutations, possibly through generation of heteroclitic neo-epitopes - but these studies are likely to take several more months and we would request that we keep these data for a later manuscript. Therefore, we do not know the full answer to the Reviewer's question here. To address it, we have added the following text on **page 21** of the **Discussion**:

We also show that APOBEC3B-driven mutagenesis generates a genomic library of heteroclitic neoantigens that, while not expressed within the *in situ* tumor, can induce anti-tumor immunity through activation of reactive anti-tumor T cells, thereby significantly enlarging the scope of available vaccine targets. **Studies are currently underway to determine whether APOBEC3B-mediated mutagenesis is uniquely able to induce enhanced immunogenicity of tumor cell vaccines or whether other mutagenic compounds/processes – such as chemotherapy or radiation therapy – can have similar effects.**

Minor comments:

1. *In the abstract and introduction please list the specific tumour types that*

simultaneously showed increased resistance to chemotherapy and simultaneously heightened sensitivity to immune checkpoint blockade.

We have added the following to the **Abstract**:

Here, we show that overexpression of APOBEC3B in tumors increased resistance to chemotherapy, but simultaneously heightened sensitivity to immune checkpoint blockade **in a murine model of melanoma.**

And to the **Introduction**:

We show here that overexpression of APOBEC3B in tumors both increased their ability to evade chemotherapy but also simultaneously conferred significantly heightened sensitivity to immunotherapy with ICB, such that tumor cures were achieved in an otherwise very poorly immunogenic **murine melanoma** tumor model.

2. Please define the HSVtk suicide gene on page 7 of the results section.

We have added the following to the **Results** on **page 7**:

B16 melanomas stably expressing the HSVtk (**Herpes Simplex Thymidine kinase**) suicide gene and....

3. In Figure 2b and c, B16-APOBEC3BActive/anti-PD1 Treating B16 and B16-APOBEC3BActive/anti-PD1 Treating TC2 are difficult to tell apart. Please change the color or shape to make the figure clearer.

We have altered the indicators on these Figures as requested by the Reviewer.

4. Please make sure to add how many times each experiment was repeated to the figures.

We have added the number of repeats to the Figure Legends in each case.

5. Is there a significant change in Fig1B between APOBEC3BACTIVE/GCV/aCTLA4 and APOBEC3BINACTIVE/GCV/aCTLA4? Most of the stats is omitted in Figure1A. Please add this to the figure and text.

We have added the stats to the Figure and to the text.

6. Please add the P-values in the text whenever comparisons are made.

We have added p values where comparisons are made.

7. The authors need to be more careful inflating the clinical potential from their data - I

imagine a whole tumour cell vaccine approach, with upregulation of abopec, would be highly challenging/controversial. They should acknowledge a lot more pre-clinical work would be needed on this, rather than immediate translation to humans.

The Reviewer's point is well taken. We do believe that translation of an APOBEC3B-modified cell vaccine, along with immune checkpoint blockade, is potentially translatable because the vaccine cells will be non-viable - as was the case in our therapy experiments here. However, the proposal to transfer APOBEC directly into tumors to induce enhanced immunogenicity through the generation of heteroclitic neo-epitopes would be highly controversial. (If attempted it would require co-expression of a cytotoxic gene to ensure eradication of any APOBEC3B-mutated cells which could, as a result, become more malignant). Therefore, as requested by the Reviewer, we have added the following text to the **Discussion** on **page 21**:

Importantly, this Heteroclitic Epitope Activated Therapy (HEAT) will contain multiple neoepitopes of which different subsets will also be heteroclitic for each individual patient, both dispensing with the need to identify patient specific neoepitopes, and overcoming the need for pre-existing targetable mutations in the cancer genome. **Nonetheless, extensive additional pre-clinical studies will have to be conducted to assess the potential for autoimmune induction using such a vaccine approach. In addition, the concept of direct transfer of APOBEC3B into live tumor cells to induce immunogenicity through formation of novel heteroclitic neo-epitopes cannot yet be considered as a clinical option due to the risks of inducing further tumor promoting mutations.**

Reviewer #3:

The authors in this manuscript demonstrate APOBEC3B mediated immune activation in B16 melanoma and GL261 glioma mice models using APOBEC3B activated vaccination strategy in combination with immune checkpoint inhibitors of T cells. The data in these two mouse models clearly suggest APOBEC3B associated tumor regression in combination with immune checkpoint inhibitors specifically associated with the APOBEC3B vaccine. Authors further demonstrate that the APOBEC3 vaccine effect in relation to neoepitopes resulted from APOBEC3B mutational signature, especially with CSDE1 minimal epitope.

However, authors' claim of CSDE1 epitope as a heteroclitic neoepitope requires further explanation/exploration. The IFN-gamma secretion by splenocyte from APOBEC3B-ACTIVE treated mice in the presence of PD1-checkpoint inhibitor when stimulated with wild type epitopes, expressed by B16 cells, is similar to B16 cells expressing no epitopes (Figure 4D). Thus, the activation of T cells could be associated with additional neoepitopes reactive T cells and not with wild type minimal epitopes reactive to T cell activated by CSDE1 neoepitope as claimed by the authors. This is also evident from the data in Figure 5D and E, that CSDE1 (neoepitope) seem to play a minor role in the tumor rejection capacity compared to the complete APOBEC3B-ACTIVE vaccine. If CSDE1 was the driver for tumor recognition (serving as a heteroclitic epitope) then it must have generated an effect similar to the complete vaccine (APOBEC3B-ACTIVE).*

Thus, the role of CSDE1 such be clarified and the conclusion be modified. The authors further validate the APOBEC3B mediated vaccine approach on human tumor samples, which can potentially be translated to clinical application. Altogether, this manuscript provide detailed and effective approach for cancer immunotherapy using APOBEC3B modification.

We apologize for over-emphasizing the role of the APOBEC3B-induced CSDE1 mutation in the protective effect of the B16-APOBEC3B^{ACTIVE} vaccine. We completely agree with the Reviewer that it is highly likely that the full immunizing potential of the APOBEC3B-modified vaccine cells is contributed by multiple mutated epitopes within the vaccine and not just the CSDE1 neo-epitope. Our intention in the manuscript was to identify one heteroclitic neo-epitope within the vaccine as an example of the mechanism by which the vaccine is working. We fully agree with the Reviewer that the full immunogenicity of the vaccine is likely to be contributed by multiple additional neo-epitopes induced by the mutagenic activity of APOBEC3B (as is indeed shown in **Figures 5D&E**). To address this point, we have added the following text to **page 19** of the **Discussion**:

B16 cell lysates expressing the APOBEC3B-altered CSDE1 peptide, but not the wild-type peptide, enhanced survival of tumor bearing mice when combined with ICB, although less effectively than the APOBEC3B-modified cell vaccine, highlighting the advantage of a wide spectrum library approach⁵⁸. Thus, we have shown that mutated CSDE1 represents at least one of the directly APOBEC3B-mutated genes which can contribute to the immunizing potential of the B16 APOBEC3B^{ACTIVE} vaccine (**Fig.5**). However, our data also clearly show that the full immunizing potential of the B16 APOBEC3B^{ACTIVE} vaccine is not recapitulated solely by mutated CSDE1 (**Figs.5D&E**). Therefore, it is highly probable that additional mutations – both within the core APOBEC3B-mediated mutational load, as well as within the subsequent mutational amplification steps – also contribute to the protective effects seen with the vaccine.

Reviewers' comments:

Reviewer #1 (Remarks to the Author):

The changes in the paper are responsive to the comments and add value.

With regard to the similar comments by rev #1, point #1 and reviewer #2, point #7: The discussion was altered appropriately. There is still the implication in the introduction, however, that using an apobec tumor in a patient might be a good idea, ie, "opening a path to clinical translation!".

lines 97-107; and especially 103-105:

"Finally, we demonstrated that human tumors can be modified through APOBEC3B expression, leading to enhanced T cell recognition, thereby opening the path to clinical translation."

I think modification here would be needed as well to point out this path is a vaccine approach.

Regarding rev #1 comment #2, the explanation is clear. Could a single line be added to the relevant place in the paper adding this point?

Reviewer #2 (Remarks to the Author):

The authors have tried to address the comments of the reviewers, however several comments have not been sufficiently addressed.

Comment 2 of Reviewer 2:

The authors do not sufficiently address comment 2 of Reviewer 2. Their explanation does not clarify how out of 1,000,000+ mutations, that are supposedly generated by APOBEC3B, only 244 contain the classical APOBEC mutational signature/motif.

The authors hypothesize that "relatively few core APOBEC3B-induced mutations will actually promote a greatly expanded and amplified profile of mutations both within any given cell, and across populations of cells." However, they don't provide any evidence. It is not convincing that only 244 mutations were created by APOBEC. What are these accessory mutations? Can the authors clarify the mutational signatures of these accessory mutations? Could these mutations be sequencing artefacts or inferior DNA quality?

The authors' following explanation is unsatisfactory "We hypothesize that the initial over-expression of APOBEC3B induces a core set of mutations, carrying the classical APOBEC3B mutational signature/motif, and often as a set of strand coordinated mutations (kategeis)⁵¹. With time, these core mutations, affect pathways in the cell which will, themselves, lead to further cascades of mutations throughout the genome. With time, the overall mutational activity seen in the cell population will reflect the sum of the initial core APOBEC3B mutations/kategeis, along with an amplification of the mutational burden through additional cascades of mutation resulting from the phenotypic effects of the core mutations. This hypothesis is consistent with the observed mutational activity of APOBEC in humans, where APOBEC mutagenesis has been shown to induce mutations in cancer driver genes in several different cancer types, which then drive the transition to an increasing malignant phenotype through additional mutation^{9, 26, 40, 52, 53, 54.} "

There is currently no evidence supporting this hypothesis, even though this aspect is central to their experimental model. The authors need to provide experimental evidence for their 'core and cascade mutations' hypothesis. Again could the authors address:

-what are these accessory mutations?

-Can the authors clarify the mutational signatures of these accessory mutations?

-Could these mutations be sequencing artefacts or inferior DNA quality?
-How many mutations are in the APOBEC motif in the APOBECinactive sample? Couldn't one expect a proportion of mutations to be in the APOBEC motif by chance? Can the authors directly compare how many mutations are in the APOBEC motif between both conditions?

Reviewer 2 minor comment 4:

-Instead of showing representative experiments, the authors would need to show averages and error bars of the aggregated experiments.

Reviewer #3 (Remarks to the Author):

In the revised version the authors has successfully addressed my previous concerns/comments

Response to Reviewers:

Reviewer #1:

The changes in the paper are responsive to the comments and add value.

We thank the Reviewer for these comments.

With regard to the similar comments by rev #1, point #1 and reviewer #2, point #7: The discussion was altered appropriately. There is still the implication in the introduction, however, that using an apobec tumor in a patient might be a good idea, ie, "opening a path to clinical translation!". lines 97-107; and especially 103-105:

"Finally, we demonstrated that human tumors can be modified through APOBEC3B expression, leading to enhanced T cell recognition, thereby opening the path to clinical translation."

I think modification here would be needed as well to point out this path is a vaccine approach. Regarding rev #1 comment #2, the explanation is clear. Could a single line be added to the relevant place in the paper adding this point?

In response to the Reviewer's concern, we have added the following text to the **Introduction** on **page 6**:

Finally, we demonstrated that human tumors can be modified through APOBEC3B expression, leading to enhanced T cell recognition, thereby opening the path to clinical translation **but only in a tumor cell vaccine setting, in which the chances of inducing any negative effects through APOBEC3B-mediated induction of increased malignancy would be abrogated by use of irradiated cell vaccines.**

Reviewer #2:

Comment 2 of Reviewer 2:

The authors do not sufficiently address comment 2 of Reviewer 2. Their explanation does not clarify how out of 1,000,000+ mutations, that are supposedly generated by APOBEC3B, only 244 contain the classical APOBEC mutational signature/motif.

We apologize for our lack of clarity here. Our NGS identified a total of ~68,000 mutations with the classical APOBEC mutational motif (not only 244). Of these ~68,000 mutations, 244 resulted in predicted amino acid changes in expressed proteins. It is these 244, predicted APOBEC-induced, coding mutations that we screened for neo-antigen discovery, from which our identification of the CSDE1* neo-epitope resulted. To clarify this, we have added the following text to **page 10** in the **Results**:

Of the ~1,000,000 total mutations detected by NGS in the B16-APOBEC3B^{ACTIVE} cells, about 68,000 contained the classical APOBEC mutational signature. Of these ~68,000 mutations, 244 resulted in predicted amino acid changes in expressed proteins through C to T or G to A transitions leading to missense mutations consistent with APOBEC3B mutational activity (Table S1)³⁵. These mutations were converted into 21-amino acid sequences, with 10 amino acids flanking the mutational site (Table S2).

The authors hypothesize that “relatively few core APOBEC3B-induced mutations will actually promote a greatly expanded and amplified profile of mutations both within any given cell, and across populations of cells.” However, they don’t provide any evidence. It is not convincing that only 244 mutations were created by APOBEC. What are these accessory mutations? Can the authors clarify the mutational signatures of these accessory mutations? Could these mutations be sequencing artefacts or inferior DNA quality?

The authors’ following explanation is unsatisfactory “We hypothesize that the initial over-expression of APOBEC3B induces a core set of mutations, carrying the classical APOBEC3B mutational signature/motif, and often as a set of strand coordinated mutations (kategeis)⁵¹. With time, these core mutations, affect pathways in the cell which will, themselves, lead to further cascades of mutations throughout the genome. With time, the overall mutational activity seen in the cell population will reflect the sum of the initial core APOBEC3B mutations/kategeis, along with an amplification of the mutational burden through additional cascades of mutation resulting from the phenotypic effects of the core mutations. This hypothesis is consistent with the observed mutational activity of APOBEC in humans, where APOBEC mutagenesis has been shown to induce mutations in cancer driver genes in several different cancer types, which then drive the transition to an increasing malignant phenotype through additional mutation^{9, 26, 40, 52, 53, 54.}”

There is currently no evidence supporting this hypothesis, even though this aspect is central to their experimental model. The authors need to provide experimental evidence for their ‘core and cascade mutations’ hypothesis.

The experimental evidence that we have to support our proposed ‘core and cascade’ model is as follows:

Our studies reported in the current manuscript identified an APOBEC-induced mutation in the *csde1* gene (at position 669) as inducing a novel, potentially immunogenic, heteroclitic neo-epitope. The B16-APOBEC3B^{INACTIVE} cells contain the *csde1* gene sequence of ATGAGCTTTGATCCA. The B16-APOBEC3B^{ACTIVE} cells contain the mutated *csde1** sequence of ATGAGCTTTGATTICA. The B16-APOBEC3B^{INACTIVE} cells contain undetectable levels of this mutated gene. We show, clearly we believe, that this APOBEC-induced mutation translates into a novel, potentially immunogenic, heteroclitic neo-epitope which contributes to the immunogenicity of the APOBEC-vaccine. This is the central focus of our manuscript.

In ongoing studies, we have further investigated the role of APOBEC activity on induction of the *csde1* mutation. We have observed that, 21 days following transduction of B16 cells with the APOBEC3B over-expressing vector, there are four separate mutations in the *csde1* gene - at positions 669 (amino acid position 5, coding change P to S), 911 (amino acid position 82, coding change P to S), 1792 (amino acid position 375, coding change H to T) and 2496 (amino acid position 610, coding change P to S).

A time course analysis of the induction of these different mutations within the *csde1* gene showed that mutations at positions 669, 911 and 2496 were induced to levels of >20% in the total cell population within 48 hours of transduction with the pBABE-Hygro APOBEC3B^{ACTIVE} vector, but were undetectable in cells transduced with the pBABE-Hygro APOBEC3B^{INACTIVE} vector. In contrast, the mutation at position 1792 was not

evident at any detectable levels at 48, 72, 96 or 120 hours post transduction with the pBABE-Hygro APOBEC3B^{ACTIVE} vector. However, the 1792 mutation was detected in the B16-APOBEC3B^{ACTIVE} cells at 21 days post transduction in up to 80% of the cell population. The 1792 mutation was not detectable in the B16-APOBEC3B^{INACTIVE} cells at any point including even at 21 days post transfection.

As we report in our current manuscript, we have observed that over-expression of APOBEC3B is toxic in that elevated levels of APOBEC3B are seen up to 72 hours post-transfection/transduction and then return to similar levels to that seen in parental unmodified cells. We believe this is because mutagenesis by APOBEC3B is tolerable to the cell up to a certain threshold, and then in cells where critical mutations are induced, this can be lethal. Consistent with this hypothesis, short term expression of APOBEC3B^{ACTIVE}, but not the catalytically inactive mutant APOBEC3B^{INACTIVE}, led to significant levels of cell killing within between 72 and 96hrs post transduction, data that we added in response to the first round of review as **Figure S4**.

Our Core and Cascade model is based upon these observations described above. That is, we observed induction of three mutations within the *csde1* gene which were detectable at levels of >20% in the whole cell population within 48hrs post transduction - during which time APOBEC3B expression was readily detected; all three of which had an APOBEC mutational signature; and all three of which were undetectable in B16 cells transduced with the APOBEC3B^{INACTIVE} vector. In contrast, we also detected one further mutation in the *csde1* gene in APOBEC3B^{ACTIVE} cells - but only by day 21 post transduction, and not at any point up to 120 hours post transduction at which time we could no longer detect APOBEC3B over-expression in the cells.

We acknowledge absolutely that these observations do not prove the overall model, nor do they identify the mutational mechanisms involved in early APOBEC activity followed by resultant mutagenesis through additional pathways. In this respect, our sequencing studies have identified candidate genes mutated in the APOBEC3B^{ACTIVE} cells that carry an APOBEC mutational signature and whose mutation may then drive further mutation in these cells (such as genes involved in chromosome structure regulation and DNA repair, including *smug1*). In order to confirm, or refute, such a model definitively, we would have to perform NGS, RNA seq and pathway analysis on B16 cells transduced with the pBABE-Hygro APOBEC3B^{ACTIVE} vector, or with the APOBEC3B^{INACTIVE} vector, at sequential time points (0, 48, 120, 7d, 14d 21d) to identify both the sequentially developing full mutational burden, and the activation of primary and secondary mutagenic pathways, in these cells with time. We do plan to carry out studies such as these but only on our candidate human vaccine cells which would form the basis of our proposed clinical trial of APOBEC3B-mutated human glioblastoma cellular vaccines. To do these NGS/RNAseq studies on our B16 cells to answer the Reviewer's point here would be expensive and would take some considerable time. In this respect, we would argue rather strongly that characterization of these mechanisms, whilst of great interest, do not form the central focus of the current manuscript – which is that APOBEC3B over-expression can drive mutation in tumor cells leading to the generation of potentially immunogenic novel heteroclitic neo-epitopes which can be exploited as powerful cancer vaccines.

We would greatly prefer to keep these data for a further manuscript which directly addresses the mechanisms of APOBEC mutagenesis (especially within the *csde1*

gene). If this is acceptable to the Reviewer, we have added the following text to the revised manuscript on **pages 17/18** to justify our introduction of the Core and Cascade model.

Our NGS revealed more than a million overall mutations in the B16 APOBEC3B^{ACTIVE} cells compared to control cells (**Fig.S2B**). Of these, ~68,000 contained the classical APOBEC signature/motif. In ongoing studies of mutagenesis of the *csde1* gene, we have observed induction of three mutations within the *csde1* gene which are detectable at levels of >20% in the whole cell population within 48hrs post transduction, during which time APOBEC3B expression was still readily detected in B16-APOBEC3B^{ACTIVE} cells. All three mutations have an APOBEC mutational signature, and all three were undetectable in B16 cells transduced with the APOBEC3B^{INACTIVE} vector. In contrast, we also detected one further mutation in the *csde1* gene in APOBEC3B^{ACTIVE} cells. However, this fourth mutation was not detectable at any point up to 120 hours post transduction - by which time we could no longer detect APOBEC3B over-expression in the cells. In contrast, this mutation was present in the population at day 21 post transduction, suggesting that it was generated subsequent to the direct mutagenic activity of APOBEC expression. None of these four mutations were detected in the B16 APOBEC3B^{INACTIVE} cells at any time points following transduction. Based on these preliminary observations, we hypothesize that the initial over-expression of APOBEC3B induces a core set of mutations, carrying the classical APOBEC3B mutational signature/motif, and often as a set of strand coordinated mutations (kategeis)⁵¹. With time, these core mutations, affect pathways in the cell which will, themselves, lead to further cascades of mutations throughout the genome. With time, the overall mutational activity seen in the cell population will reflect the sum of the initial core APOBEC3B mutations/kategeis, along with an amplification of the mutational burden through additional cascades of mutation resulting from the phenotypic effects of the core mutations. This hypothesis is consistent with the observed mutational activity of APOBEC in humans, where APOBEC mutagenesis has been shown to induce mutations in cancer driver genes in several different cancer types, which then drive the transition to an increasing malignant phenotype through additional mutation^{9, 26, 40, 52, 53, 54}. Moreover, APOBEC activity has been shown to drive branched evolution through the acquisition of sub-clonal mutations during the evolutionary progression of a variety of human cancers^{55, 56, 57} which is consistent with the presence of the sub-clonal, APOBEC signature mutations that we detected in the B16 APOBEC3B^{ACTIVE} vaccine population (**Fig. S2B**).

.... Again could the authors address:

-what are these accessory mutations? -Can the authors clarify the mutational signatures of these accessory mutations?

We have provided an analysis of these accessory mutations in **Supplemental Figure S2B**.

-Could these mutations be sequencing artefacts or inferior DNA quality?

All of the sequencing samples on which we report passed the quality control of our Core facility and were run together, so we do not believe that they are artefactual.

-How many mutations are in the APOBEC motif in the APOBECinactive sample? Couldn't one expect a proportion of mutations to be in the APOBEC motif by chance? Can the authors directly compare how many mutations are in the APOBEC motif between both conditions?

To answer the Reviewer's comments here, we analyzed the frequency of mutation of candidate APOBEC signature motifs which were mutated in the B16-APOBEC3B^{ACTIVE} cells (hA3B-WT) compared to the parental B16 cells (21 days post transduction with the APOBEC3B^{ACTIVE} vector) (see Figure below). Of those motifs which were mutated in the B16-APOBEC3B^{ACTIVE} cells, we were unable to detect any mutations in the B16-APOBEC3B^{INACTIVE} cells (hA3B-Catalytic Mutant CM).

So, at least at the level of detection of our sequencing, in those APOBEC motifs mutated in the B16-APOBEC3B^{ACTIVE} condition, we did not observe detectable mutation in the B16-APOBEC3B^{INACTIVE} condition. We agree with the Reviewer that APOBEC motifs could be mutated by chance, or by other mutagenic pathways, in the B16-APOBEC3B^{ACTIVE} cells. However, we did not detect such chance mutations in the B16-APOBEC3B^{INACTIVE} condition (see Figure above), suggesting that the frequency is low. These data also show that there is a very clear bias for possible APOBEC motifs in the mutations which are induced in cells transduced with the APOBEC3B^{ACTIVE} vector compared to parental, or B16-APOBEC3B^{INACTIVE}, cells. We would greatly prefer to keep these data for a further manuscript which directly addresses the mechanisms of APOBEC mutagenesis (especially within the *csde1* gene) rather than having to expand the scope and length of the current manuscript.

Reviewer 2 minor comment 4:

-Instead of showing representative experiments, the authors would need to show averages and error bars of the aggregated experiments.

We have now done this.

Reviewer #3:

In the revised version the authors has successfully addressed my previous concerns/comments

We thank the Reviewer for this comment.

REVIEWERS' COMMENTS:

Reviewer #2 (Remarks to the Author):

The authors have sufficiently addressed previous concerns.